# FedDrop: Trajectory-weighted Dropout for Efficient Federated Learning

## Abstract

Federated learning (FL) enables edge clients to train collaboratively while preserving individual's data privacy. As clients do not inherently share identical data distributions, they may disagree in the direction of parameter updates, resulting in high compute and communication costs in comparison to centralized learning. Recent advances in FL focus on reducing data transmission during training; yet they neglected the increase of computational cost that dwarfs the merit of reduced communication. To this end, we propose FedDrop, which introduces channel-wise weighted dropout layers between convolutions to accelerate training while minimizing their impact on convergence. Empirical results show that FedDrop can drastically reduce the amount of FLOPs required for training with a small increase in communication, and push the Pareto frontier of communication/computation trade-off further than competing FL algorithms.

## 1 Introduction

In the light of the importance of personal data and the recent strict privacy regulations, *e.g.* the General Data Protection Regulation (GDPR) of the European Union (Voigt & Von dem Bussche, 2017; Wolters, 2017; Politou et al., 2018), there is now a great amount of risk, responsibility (Edwards et al., 2016; Culnan & Williams, 2009) and technical challenges for securing private data centrally (Sun et al., 2014); it is often impractical to upload, store and use data on central servers. To this end, *federated learning* (FL) (McMahan et al., 2017; Li et al., 2019a) enables multiple edge compute devices to learn a global shared model collaboratively in a communication-efficient way without collecting their local training data. When compared against naïve decentralized SGD, *Federated averaging* (FedAvg) (McMahan et al., 2017) and subsequent FL algorithms (Li et al., 2020; Karimireddy et al., 2020) reduce the burden of data transmission by many orders of magnitude.

While these communication-efficient methods can notably alleviate the difficulties of using FL in scenarios with limited bandwidth or data-quota, they, however, entail a drastic increase in computation cost, which has rarely been addressed by previous literature on FL. It has been argued that communication is several orders of magnitude more expensive than computation on edge devices (Li et al., 2019a; Huang et al., 2013). Yet existing FL methods optimize for communication so aggressively that an iPhone 12 Pro running as a FedAvg client in our test[1] would spend at least 16 minutes on training, before even transmitting 180 MB of data, which only takes fraction of a second under the modern 5G infrastructure. The savings in communication thus are completely dwarfed by the expensive computation.

In a FL setup, models trained by the clients may disagree on the gradient direction to update the same neurons if the user data distributions are non-IID. In Figure 1, we demonstrate this effect with an initial round of FedAvg training. It turns out that clients sharing similar data distributions tend to produce similar update trajectories to the same channel, and different distributions observe disparate trajectories. From this example, it can be observed that the consequence of non-IID data is two-fold. First, a naïve averaging of client parameters may cause these conflicting signals to cancel out each other, resulting in a slow convergence of the global model. Second, neurons in a layer tend to learn distinct features, and yet they cannot learn meaningfully if these features are absent in the client's training data. Since neuron training is heavily dependent on the client's local data, it presents us an

---

[1]See Appendix F.6 for the setup.

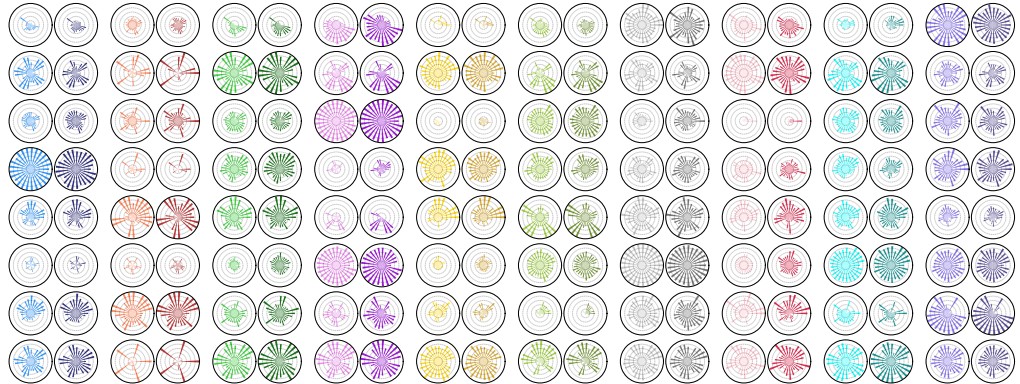

Figure 1: We synthesized 20 clients with the Fashion-MNIST training data, where every two clients received only the same class images to simulate a concept shift, and separately trained the same CNN model for 1 epoch. We present all filter parameters update magnitudes for the first 8 channels (row) in the first convolutional layer for all clients (column) after 1 round of training. The pairs of clients grouped together with same colors (one lighter and one darker) signify that they shared the same data distribution. Each polar plot contains the update magnitudes of all parameters in the same channel. The length of each ray is the magnitude of the parameter update, and the parameters are arranged radially in each plot.

opportunity: *can we concentrate training effort to neurons that are correlated to the current data distribution of the client, while paying less attention to neurons that are less relevant to the client?* To leverage this, we introduce FedDrop, which introduces SyncDrop layers to structurally sparsify client models, and on the server-side an inter-client trajectory-based optimization of the dropout probabilities used by SyncDrop to speed up model training. FedDrop brings two-fold advantages: dropped neurons can simply be skipped, and thus reduce the training FLOPs required per step; and dropout probabilities of each neuron can be tuned individually to minimize the impact on the global averaged model updates to assist convergence. Overall, as evinced by our experiments, FedDrop enables a much improved communication/computation trade-off relationship than traditional FL approaches.

Our contributions in this paper are as follows:

- We present SyncDrop layers, which are synchronous structural dropouts with adaptive keep probabilities, to reduce the computational cost required by FL.

- With SyncDrop, we formally derive the FedDrop objective that adjusts each neuron's dropout probability to improve convergence by minimizing the disparities among inter-client update trajectories. We further introduce FLOPs-based constraints to enforce sparsity per FL round, allowing the trade-off between FLOPs and communication to be tuned easily.

- Empirical results reveal that the combined method, FedDrop, attains a substantially better communication/computation trade-off in comparison to other FL methods.

## 2 RELATED WORK

**Federated learning**. Distributed machine learning has a long history of progress and success (Peteiro-Barral & Guijarro-Berdiñas, 2013; Li et al., 2014), yet it mainly focuses on training with IID data. The Federated Learning (FL) paradigm and the Federated Averaging algorithm (FedAvg) initially introduced by McMahan et al. (2017) allow clients to train collaboratively without sharing the private data in a communication-efficient manner, To further tackle data heterogeneity, FedProx (Li et al., 2020) introduces new regularizations, and SCAFFOLD (Karimireddy et al., 2020) presents control variates to account for client drifts and reduce the inter-client variance. While being effective at reducing communication, the above methods neglected the computational costs associated with the training process.

**Computation *vs.* communication during training**. There are a few precursory methods that focus on the joint optimization of computation and communication costs during training. Caldas et al. (2018a) introduced federated dropout, which prunes parameters following a uniform random distribution, whereas PruneFL (Jiang et al., 2019) proposes a greedy gradient-magnitude-based unstructured pruning. In each FL round, both methods produce a shared pruned model with fine-grained sparsity for all clients. Such unstructured sparsity is difficult to utilize for training acceleration, and a shared global model cannot exploit the data distribution of individual clients. Adaptive federated dropout (Bouacida et al., 2020) partially addresses the latter issue by allowing each client to select a sub-model to join training. FjORD (Horvath et al., 2021) introduces ordered dropout to tackle the problem of system heterogeneity in FL with sub-model training that dynamically changes model capabilities using uniform sampling. FedGKT (He et al., 2020) transmits shallow layer activations to offload training of subsequent layers to the server. However, the privacy implications were not well explored, and the trained models were substantially larger, which may limit their inference speed on edge devices. It is noteworthy that FedDrop differs from these approaches as it takes into account the non-IID client data distributions in a FL setting, and computes inter-client channel selection decisions that would minimize the impact on convergence.

**Dropout algorithms in centralized training**. Dropout (Hinton et al., 2012; Srivastava et al., 2014) may improve neural network training by reducing the overfitting effect and allow it to generalize better. It is done by randomly setting parts of the connections or weights to zero in a network, and scaling the remaining values accordingly. Since its advent, there have been an increasing interest in applying dropout in a structural fashion (Huang et al., 2015; Ghiasi et al., 2018; Hou & Wang, 2019). Nonetheless, the above methods missed the opportunity to explore the implications of a structural dropout on the FLOPs consumed by training or inference.

**Structural pruning**. A closely related topic is structural neuron pruning/selection for accelerated inference. These work propose new ways to extract a smaller accurate network from the original (He et al., 2018; Wu et al., 2019; Li et al., 2019b; Herrmann et al., 2020), and can even do so dynamically with an input-dependent policy (Gao et al., 2019; Hua et al., 2019; Wang et al., 2020c).

## 3 THE FEDDROP METHOD

### 3.1 HIGH-LEVEL OVERVIEW

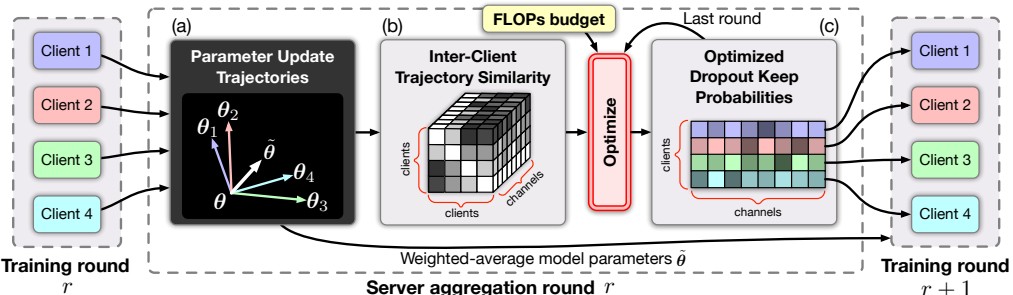

Figure 2: A high-level overview of FedDrop using FedAvg as the base algorithm. Here, we use 4 training clients for illustration. During each round of server averaging, FedDrop carries out dropout probability optimization to encourage collaboration between sparse clients for the next training round. Intuitively, for each neuron, if clients agree upon an update direction they would use larger keep probabilities, and a disagreement results in lower probabilities. Clients can thus focus training effort to their specialized neurons.

FedDrop complements existing FL methods by adding new client- and server-side components. During client training, convolutional layers are sparsified by interleaving them with channel-wise dropout layers, namely the SyncDrop layers. Section 3.3 discusses in depth how SyncDrop layers compute dropout decisions. Additional optimization stages during server aggregation are added to encourage collaboration between sparse clients. Figure 2 shows how FedDrop extends traditional FedAvg (McMahan et al., 2017). After each round of client training, the server begins by identifying

parameter update directions of each client from the previous round, and computes a cross-client trajectory similarity matrix for each channel neuron. This is then followed by an optimization stage that iteratively minimizes the trajectory disparities by tuning dropout keep probabilities for each channel neuron on each client for the next training round (Section 3.6). This process also takes into consideration the FLOPs budget constraints to sparsify client models (Section 3.4). Finally, following FedAvg, the server broadcasts the weighted-average model parameters to all training clients in the new round, and finally sends the optimized probabilities to the respective clients.

## 3.2 Preliminaries and Definitions

FedDrop complements most FL algorithms. In this paper, we focus on its use on FedAvg (McMahan et al., 2017). We assume the training loss function of a client $c \in \mathbb{C}$ to be $\ell_c(\boldsymbol{\theta}_c)$, where $\boldsymbol{\theta}_c$ comprises the parameters $\{\boldsymbol{\theta}_c^{[l]} : l \in L\}$ of all layers in the model of client $c$.

In each FedAvg training round $r$, clients begin by training on the loss function, with initial parameters $\boldsymbol{\theta}^{(r)}$ received from the server for this round:

$$\boldsymbol{\theta}_c^{(r+1)} = \mathsf{SGD}_c\big(\ell_c, \boldsymbol{\theta}^{(r)}, \eta, E\big). \tag{1}$$

Here $\mathsf{SGD}_c$ indicates that client $c$ carries out stochastic gradient descent (SGD) on $\ell_c(\boldsymbol{\theta}_c^{(r)})$ locally, and it uses a learning rate $\eta$ for $E$ epochs. The FedAvg server then aggregates client model parameters after the $r^{\text{th}}$ training round, by taking the weighted average of them:

$$\boldsymbol{\theta}^{(r+1)} = \sum_{c \in \mathbb{C}} \lambda_c \boldsymbol{\theta}_c^{(r+1)}, \tag{2}$$

where $\lambda_c$ is the weight of client $c$ and is proportional to the size of its training set $|\mathbb{D}_c|$ with $\sum_{c \in \mathbb{C}} \lambda_c = 1$. Finally, the $(r+1)^{\text{th}}$ training round starts by repeating the above procedure.

## 3.3 Synchronized Dropout

To induce sparsity in a stochastic manner, and subsequently introduce training correlation across clients, we designed a threshold-based dropout layer, SyncDrop, to synchronize dropout decisions across multiple clients.

Initially for all sampled clients $c \in \mathbf{C} \subseteq \mathbb{C}$, the server provides them with their corresponding dropout keep probabilities $\mathbf{p}_c \in [0, 1]^{\mathbb{N}}$, and we assume $\mathbf{p} \in [0, 1]^{\mathbf{C} \times \mathbb{N}}$ to be the concatenated probabilites from all sampled clients. Each element $\mathbf{p}_c^n$ in $\mathbf{p}$ denotes the keep probability of the channel neuron $n \in \mathbb{N}$ in client $c \in \mathbb{C}$. We additionally use $\mathbf{p}_c^{[l]}$, a slice of $\mathbf{p}_c$, to indicate the probabilities of all channels in the $l^{\text{th}}$ layer. During training, the $l^{\text{th}}$ layer samples $\mathbf{t}_n^{[l]}$ from a uniform distribution $\mathcal{U}(0, 1)$ for all channel neurons $n$ in the layer, where $\mathbf{t}_n^{[l]}$ is shared across clients by using the same random seed. If $\mathbf{p}_c^n < \mathbf{t}_n^{[l]}$, a channel $n$ in client $c$ is dropped, *i.e.* we set the entire channel map to zero; otherwise, the channel values are scaled by $1/\mathbf{p}_c^n$ during training[2]. Formally, for each client $c$, the $l^{\text{th}}$ dropout layer computes for the input $\mathbf{x}^{[l]}$:

$$\mathsf{drop}\big(\mathbf{x}^{[l]}, \mathbf{p}_c^{[l]}\big) \triangleq \mathbf{d}_c^{[l]} \circ \mathbf{x}^{[l]} \circ \mathbf{p}_c^{[l] \circ -1}, \text{where } \mathbf{d}_c^{[l]} \triangleq \mathbb{1}[\mathbf{t}^{[l]} < \mathbf{p}_c^{[l]}]. \tag{3}$$

Here, $\mathbb{1}[\mathbf{z}]$ is the (element-wise) indicator function, which is equal to 1 when the condition $\mathbf{z}$ is met and 0 otherwise, $\circ$ refers to the element-wise product, the term $\mathbf{p}_c^{[l] \circ -1}$ represents the element-wise inverse of $\mathbf{p}_c^{[l]}$, and finally all elements of $\mathbf{t}^{[l]}$ are independently sampled from $\mathcal{U}(0, 1)$ and shared across clients. Figure 3a provides a high-level overview of the procedure described above.

From the local perspective of a client $c$, the dropout decision $\mathbf{d}_c^{[l]}$ is equivalent to the Bernoulli distributions $\mathcal{B}(\mathbf{p}_c^{[l]})$. An important distinction from independent Bernoulli distributions is that the distribution of $\mathbf{d}_c^{[l]}$ is correlated across clients $c \in \mathbb{C}$. Given a pair of clients $(i, j)$ for channel $n$:

$$\mathsf{E}_{t \sim \mathcal{U}(0,1)}\big[\mathbf{d}_i^n \mathbf{d}_j^n\big] = \int_0^1 \mathbb{1}[t < \mathbf{p}_i^n] \cdot \mathbb{1}[t < \mathbf{p}_j^n] \mathrm{d}t = \min\big(\mathbf{p}_i^n, \mathbf{p}_j^n\big). \tag{4}$$

This enables us to adjust the correlation of the same channel neurons between any pairs of clients, and Section 3.6 makes use of this property to minimize the model update disparities across all clients.

---

[2]This is known as "inverted dropout" and implemented by PyTorch (2021) and TensorFlow (2021). Figure 7 provides ablation results on scaling for dropout to justify the design choice.

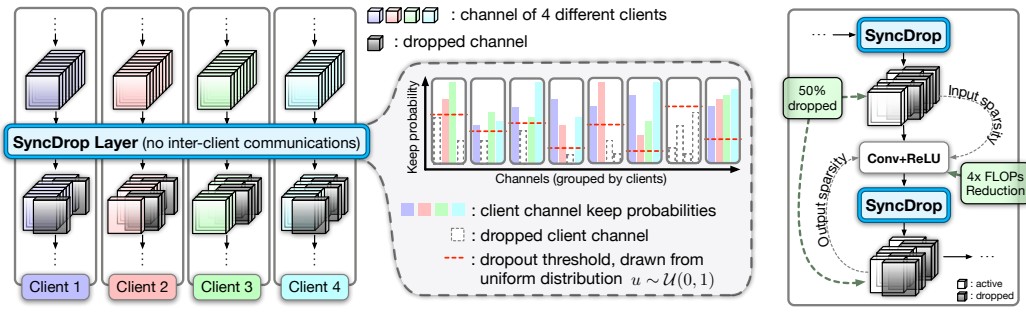

| (a) The SyncDrop layer. | (b) Sparse convolution. |

Figure 3: A high-level overview of the SyncDrop layer. (a) For each channel across all clients in a forward pass, we draw $u \sim \mathcal{U}(0,1)$ from the uniform distribution. If the keep probability ($1 -$ dropout probability) of a channel in a client is less than $u$, then the channel is dropped from the model. Note that no inter-client communications are required, as the clients are synchronized by sharing an initial random seed. (b) When a convolutional layer is sandwiched between two SyncDrop layers, it takes the advantage of both the input- and output-side sparsities. As an example, here dropping 50% of both input and output channels can result in a $4\times$ FLOPs reduction.

We define $\hat{f}$ to be the sparsified variant of the original model $f$, where SyncDrop layers are placed immediately after each convolutional layer with ReLU activation (Figure 3b). This enables convolutional layers to be doubly accelerated by taking advantage of sparsities at both ends and skipping dropped channels in input/output feature maps. Recalling the client model loss function $\ell_c$, we additionally use $\hat{\ell}_c$ to denote the accelerated variant of the sparse model $\hat{f}$ with probabilities $\mathbf{p}_c$.

It is noteworthy that active channels are sampled for each local training mini-batch. In addition, the synchronization of dropout decisions is carried out with identical random seeds for all clients, and thus it eliminates the need to communicate between clients. Finally, for inference, one may choose to enable the SyncDrop layers for speed, or skip these layers entirely. In our evaluations, SyncDrop layers are disabled for improved test accuracies.

## 3.4 FLOPS CONSTRAINTS

To encourage sparsity in models and for the optimization of $\mathbf{p}$, we adopt a hyper-parameter $r \in (0, 1]$ that adjusts the FLOPs budget ratio, and define a shared global FLOPs-budget constraint $g(\mathbf{p}) \geq 0$ for all participating clients $\mathbf{C} \subseteq \mathbb{C}$ in one round, where:

$$g(\mathbf{p}) = r - \sum_{c \in \mathbf{C}} \text{flops}(\hat{f}, \mathbf{p}_c) / (|\mathbf{C}| \, \text{flops}(f)), \tag{5}$$

and the terms $\text{flops}(\hat{f}, \mathbf{p}_c)$ and $\text{flops}(f)$ respectively denote the FLOPs of a sparse model with probabilities $\mathbf{p}_c$, and the FLOPs of the dense model $f$. Moreover, this constraint can be easily modified to allow client- and layer-wise, and even heterogeneous budgets by summing over targets with finer granularity. In Appendix B, we explain how the FLOPs of a model is computed.

## 3.5 PARAMETER INITIALIZATION

At model initialization, we set all values in $\mathbf{p}$ to be a uniform constant $p$ such that $g(\mathbf{p}) = 0$. Since the training clients are sparse at initialization, the variance of each initialized parameters $\boldsymbol{\theta}_{ij}^{[l]} \in \boldsymbol{\theta}^{[l]}$ of the $l^{\text{th}}$ layer must satisfy the following condition:

**Theorem 1** (Parameter initialization). *To avoid vanishing/exploding gradients at initialization, the variance of the parameters of the $l^{\text{th}}$ layer followed by a dropout layer with keep probability $p$ should be $2p/N$, where $N = C^{[l-1]}K^{[l-1]2}$ is the size of the receptive field of each output channel.*

In this paper, we introduce a modified He initialization (He et al., 2015), that is $\theta \sim \mathcal{N}(0, \sqrt{2p/N})$. In Appendix C.1 we provide a proof of this theorem. Both Hendrycks & Gimpel (2016) and Pretorius et al. (2018) derived similar schemes for fixing parameters initialization of layers followed by dropout. Figure 7 further justifies this decision empirically by ablation of the fixed initialization.

### 3.6 Optimizing Dropout Keep Probabilities

In our experiments, we found that using a constant dropout keep probability is often detrimental to the joint optimization of computation and communication costs. To speed up training while minimizing the impact of model sparsity on global training convergence, we therefore aim to optimize the dropout keep probabilities $\mathbf{p}$ for the following objective:

$$\min_{\mathbf{p}} \mathsf{E} \left\| \sum_{c \in \mathbb{C}} \lambda_c \nabla_{\boldsymbol{\theta}_c} \hat{\ell}_c - \sum_{c \in \mathbb{C}} \lambda_c \nabla_{\boldsymbol{\theta}_c} \ell_c \right\|_2^2, \tag{6}$$

*i.e.* the mean square error between the averaged gradients of all training clients with and without SyncDrop layers. Intuitively, it aligns the parameter updates of sparse models with the original dense variants for an improved convergence behavior. As a result of the correlated dropouts across clients (4), the objective above can be minimized by adjusting all dropout keep probabilities $\mathbf{p}_c^n$ across neuron $n$ on client $c$.

As it is impractical to communicate each step of SGD across clients, FedDrop instead optimizes an alternative objective derived in the theorem below (proved in Appendix C.2) that approximately minimizes the original:

**Theorem 2** (Approximate Objective). *Assuming that for a channel neuron $n \in \mathbb{N}$ and a pair of clients $i, j \in \mathbb{C}$, we have $\hat{\mathbf{S}}_{ij}^n \triangleq \lambda_i \lambda_j \max(\mathbf{p}_i^n, \mathbf{p}_j^n) \Delta \boldsymbol{\theta}_i^n \cdot \Delta \boldsymbol{\theta}_j^n$, where $\Delta \boldsymbol{\theta}_i^n, \Delta \boldsymbol{\theta}_j^n$ are the parameter updates after a round of FL training, $\mathbf{p}_i^n, \mathbf{p}_j^n$ are the current dropout keep probabilities, and $\max(\mathbf{x}, \mathbf{y})$ represents the element-wise max between $\mathbf{x}$ and $\mathbf{y}$, the optimization objective of (6) can be approximated by:*

$$\min_{\mathbf{q}} \mathsf{obj}(\hat{\mathbf{S}}, \mathbf{q}) \triangleq \min_{\mathbf{q}} \sum_{n \in \mathbb{N}, i, j \in \mathbb{C}} \hat{\mathbf{S}}_{ij}^n / \max(\mathbf{q}_i^n, \mathbf{q}_j^n). \tag{7}$$

The optimized values $\mathbf{q}$ can subsequently be used as the new dropout keep probabilities for the next round of FL training.

It is also desirable to have a bounded optimization to avoid extreme solutions. Without a FLOPs constraint, both objectives (6) and (7) can be trivially minimized with $\mathbf{p} = \mathbf{1}$, *i.e.* when none of the channels are dropped, and Appendix C.2 provides the proofs of the following theorem:

**Theorem 3** (Optimization bound). *The objective $\min_{\mathbf{q}} \sum_{nij} \hat{\mathbf{S}}_{ij}^n / \max(\mathbf{q}_i^n, \mathbf{q}_j^n)$ is trivially minimized to $\sum_{nij} \hat{\mathbf{S}}_{ij}^n$ when $\mathbf{q}_c^n = 1$ for $\forall n \in \mathbb{N}, c \in \mathbf{C}$.*

To enforce sparsity, we therefore further constrain the optimization of $\mathbf{q}$ to be within the feasible set $g(\mathbf{p}) \geq 0$ as defined in Section 3.4. In practice on the server-side after gathering model updates from a round of sparse client training, we minimize the overall objective via gradient descent with the interior point method:

$$\min_{\mathbf{q}} \mathsf{obj}(\hat{\mathbf{S}}, \mathbf{q}) - \mu \log(g(\mathbf{q})). \tag{8}$$

The term $-\mu \log(g(\mathbf{q}))$ constrains the solution $\mathbf{q}$ to be within the feasible set $g(\mathbf{q}) \geq 0$, and the hyper-parameter $\mu$ tunes the strength of the regularization, and is kept constant at $10^{-4}$. In Appendix D, we describe the overall FedDrop algorithm in depth, and discuss the computational and memory implications of FedDrop.

## 4 Experimental Results

In this section, we present a comprehensive evaluation of both the communication and computation costs of FedDrop and compare it against other FL algorithms, namely FedAvg (McMahan et al., 2017), FedProx (Li et al., 2020), SCAFFOLD (Karimireddy et al., 2020), and Caldas et al. (2018a). We also introduce UniDrop for reference — a simple channel-wise dropout with a uniform keep probability that is kept constant as the comparison baseline.

We conduct experiments on three popular open datasets CIFAR-10 (Krizhevsky et al., 2014), Fashion-MNIST (Xiao et al., 2017) and SVHN (Netzer et al., 2011). To simulate non-IID training data distribution with a concept shift, for every class label we split the training dataset into $|\mathbb{C}|$ parts to be respectively received by the clients, where the size of all parts follow the Dirichlet distribution $\mathcal{D}_{|\mathbb{C}|}(\alpha)$ (Hsu et al., 2019; Wang et al., 2020b). For the full participation case, we used $\alpha = 0.5$, 20 clients for CIFAR-10, and 100 clients for Fashion-MNIST and SVHN in all experiments by default.

We used a 9-layer VGG-style architecture (Simonyan & Zisserman, 2015) for CIFAR-10, wheras Fashion-MNIST used a LeNet (Lecun et al., 1998) model and SVHN a smaller variant, and we provide the details of the models and datasets in Appendix E.

For a fair comparison, we performed a grid-based search on the remaining hyperparameters for baseline algorithms to find configurations that provide the best accuracy after 100 rounds of training for each dataset. The search results suggested that typically setting the batch size $B = 4$ and learning rate $\eta = 0.02$ were universally optimal for most methods under comparison. FedProx additionally introduced a regularizer $\mu$ which was also optimized within $\{0.0001, 0.001, 0.01, 0.1\}$. Finally, FedDrop introduces a global FLOPs budget per round as a ratio between $r \in (0, 1]$ that can be adjusted to steer the communication/computation trade-off as defined in (5), for which we specify the values in the experiments below.

**FLOPs *vs*. Accuracy.** We first focus on the amount of FLOPs required by the algorithms, and provide a set of comparisons across the FL methods for the same number of local epochs per round $E$, given their respective optimal hyperparameters with an unbounded communication budget (Figure 4). Here, we used a FLOPs ratio $r = 0.5$ for FedDrop. Frequent communication rounds allow the server to more frequently optimize dropout keep probabilities for FedDrop, which can notably improve convergence under the same FLOPs budget. In contrast, other methods struggle to improve their convergence speed.

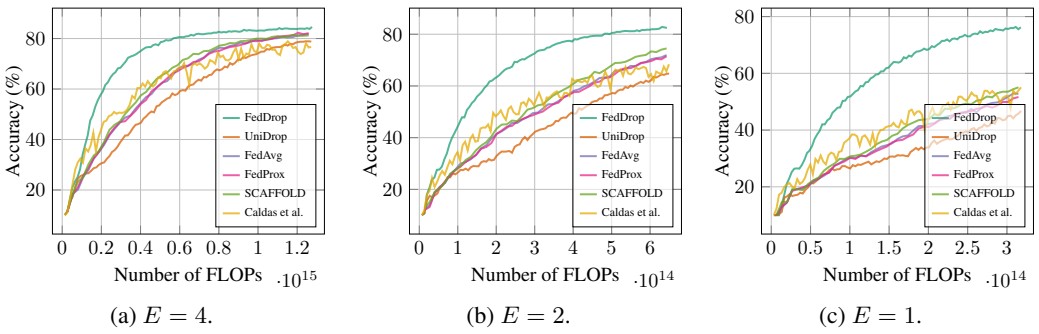

Figure 4: Comparing FL methods with the same local training epochs per round $E$ used for CIFAR-10.

**FLOPs *vs*. Communications.** To bolster the trade-off between communication and computation, we additionally conducted a hyperparameter exploration to navigate the Pareto frontiers of this trade-off relationship. The hyperparameters include the number of local epochs per round $E$ and the batch size $B$. In addition, FedDrop can search for the Pareto optimal $r$ in the set $\{0.2, 0.25, 0.333, 0.5\}$. Figure 5 shows the FLOPs/communication trade-off of the FL methods. It is notable that with decreasing FLOPs budgets, FedDrop expends minimal communications, whereas other methods may have significantly larger communication costs to reach the same target accuracy. Conversely, in Table 1, we show that FedDrop can be up to $3\times$ more efficient in terms of FLOPs under the same communication budget for the same accuracy goal.

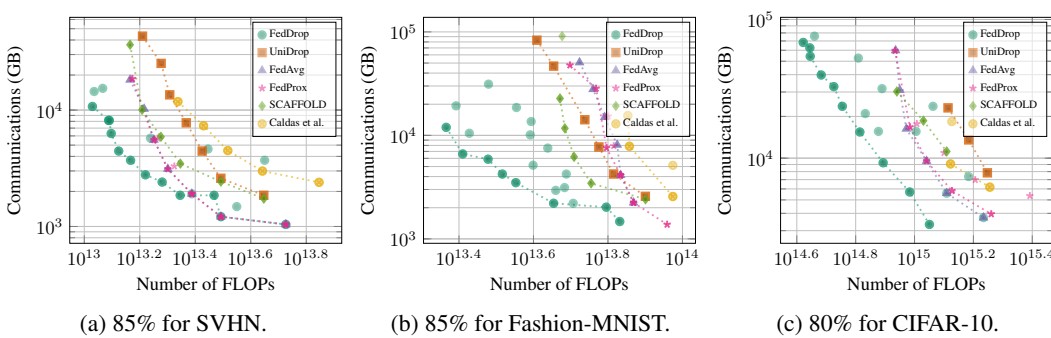

Figure 5: Comparing the FLOPs *vs*. communication trade-off across different FL methods reaching a target accuracy for a given dataset. We highlight the Pareto optimal points with dotted lines.

Table 1: Comparing the computational costs (in TFLOPs) required by FedDrop against other methods given a communications budget. Multipliers in parentheses signify the magnitude of FLOPs increase compared to FedDrop.

| Dataset | Acc. | Comm. | Methods | | | | | |
|---|---|---|---|---|---|---|---|---|
| | | | FedDrop | UniDrop | FedAvg (McMahan et al., 2017) | FedProx (Li et al., 2020) | SCAFFOLD (Karimireddy et al., 2020) | Caldas et al. (2018a) |
| CIFAR-10 | 70% | 8 GB | **391.35** | 964.12 (2.46×) | 602.63 (1.54×) | 669.59 (1.71×) | 696.37 (1.78×) | 715.19 (1.83×) |
| | | 16 GB | **244.79** | 856.99 (3.50×) | 569.15 (2.32×) | 589.24 (2.41×) | 549.06 (2.24×) | 622.48 (2.54×) |
| | 80% | 16 GB | **753.67** | 1526.52 (2.01×) | 937.42 (1.24×) | 1098.73 (1.46×) | 1071.34 (1.42×) | 1324.43 (1.76×) |
| | | 32 GB | **531.43** | 1298.88 (2.44×) | 890.55 (1.68×) | 964.21 (1.81×) | 870.47 (1.64×) | 1172.12 (2.21×) |
| Fashion-MNIST | 80% | 4 GB | **12.85** | 29.81 (2.32×) | 32.68 (2.54×) | 32.68 (2.54×) | 31.25 (2.43×) | 34.82 (2.71×) |
| | | 8 GB | **9.82** | 26.97 (2.74×) | 30.54 (3.11×) | 30.54 (3.11×) | 26.99 (2.75×) | 31.65 (3.22×) |
| | 85% | 4 GB | **32.83** | 65.30 (1.99×) | 68.19 (2.08×) | 68.19 (2.08×) | 56.83 (1.73×) | 94.78 (2.87×) |
| | | 8 GB | **25.69** | 59.62 (2.32×) | 66.77 (2.60×) | 62.51 (2.43×) | 51.14 (1.99×) | 72.02 (2.80×) |
| SVHN | 80% | 4 GB | **7.72** | 14.43 (1.87×) | 12.25 (1.59×) | 12.25 (1.59×) | 12.22 (1.58×) | 17.54 (2.27×) |
| | | 8 GB | **6.45** | 13.08 (2.02×) | 11.41 (1.77×) | 11.42 (1.77×) | 11.70 (1.81×) | 13.48 (2.09×) |
| | 85% | 4 GB | **13.30** | 26.60 (2.00×) | 19.99 (1.50×) | 19.99 (1.50×) | 22.19 (1.67×) | 43.85 (3.30×) |
| | | 8 GB | **12.23** | 23.31 (1.91×) | 17.82 (1.46×) | 17.82 (1.46×) | 16.15 (1.32×) | 26.89 (2.20×) |

**Fractional Device Participation.** To measure the performance of FedDrop in more practical distributed scenario, we scale the number of devices to 1000, while reducing the fraction of participation rate $\phi$ per round to only 1%, yielding results in Table 2. We refer readers to Appendix F.5 for more detailed trade-off results.

Table 2: Comparing the computational costs (in TFLOPs) required by FedDrop against other methods given a communications budget with 1000 devices and 1% device participation ratio ($\phi = 0.01$).

| Dataset | Acc. | Comm. | Methods | | | | |
|---|---|---|---|---|---|---|---|
| | | | FedDrop | UniDrop | FedAvg (McMahan et al., 2017) | FedProx (Li et al., 2020) | Caldas et al. (2018a) |
| CIFAR-10 | 70% | 160 GB | **122.74** | 290.98 (2.37×) | 215.07 (1.75×) | 213.46 (1.74×) | 469.64 (3.83×) |
| | | 320 GB | **99.89** | 213.96 (2.14×) | 194.50 (1.95×) | 187.74 (1.88×) | 238.93 (2.39×) |
| | 75% | 160 GB | **223.28** | 392.79 (1.76×) | 322.47 (1.44×) | 321.13 (1.44×) | 735.59 (3.29×) |
| | | 320 GB | **128.52** | 319.57 (2.49×) | 233.15 (1.81×) | 225.55 (1.75×) | 396.93 (3.09×) |
| Fashion-MNIST | 80% | 4 GB | **1.32** | 1.89 (1.43×) | 2.70 (2.05×) | 2.59 (1.96×) | 2.89 (2.19×) |
| | | 8 GB | **1.04** | 1.64 (1.58×) | 2.25 (2.16×) | 2.32 (2.23×) | 2.28 (2.19×) |
| | 85% | 4 GB | **4.83** | 6.72 (1.39×) | 6.00 (1.24×) | 6.01 (1.24×) | 7.78 (1.61×) |
| | | 8 GB | **2.77** | 5.59 (2.02×) | 5.54 (2.01×) | 5.47 (1.97×) | 6.50 (2.35×) |
| SVHN | 80% | 2 GB | **0.66** | 1.11 (1.68×) | 1.05 (1.59×) | 1.04 (1.58×) | 1.39 (2.11×) |
| | | 4 GB | **0.56** | 0.99 (1.77×) | 0.95 (1.70×) | 0.91 (1.62×) | 1.14 (2.04×) |
| | 85% | 2 GB | **1.36** | 2.34 (1.72×) | 1.65 (1.21×) | 1.60 (1.18×) | 3.10 (2.28×) |
| | | 4 GB | **0.94** | 2.17 (2.31×) | 1.40 (1.49×) | 1.37 (1.46×) | 2.50 (2.66×) |

**Dataset Distribution Imbalance.** To investigate how different data distributions affect the performance of FedDrop rigorously, we provide a sweep of $\alpha \in \{5, 0.5, 0.05\}$ for the CIFAR-10 dataset splits $\mathcal{D}_{|\mathbb{C}|}(\alpha)$ to simulate increasing degrees of distribution imbalance in Figure 6. It shows that FedDrop is the best method at maintaining accuracy with distribution imbalance *w.r.t.* the FLOPs budget. We believe SCAFFOLD may be unsuited for larger models as it diverged under these scenarios.

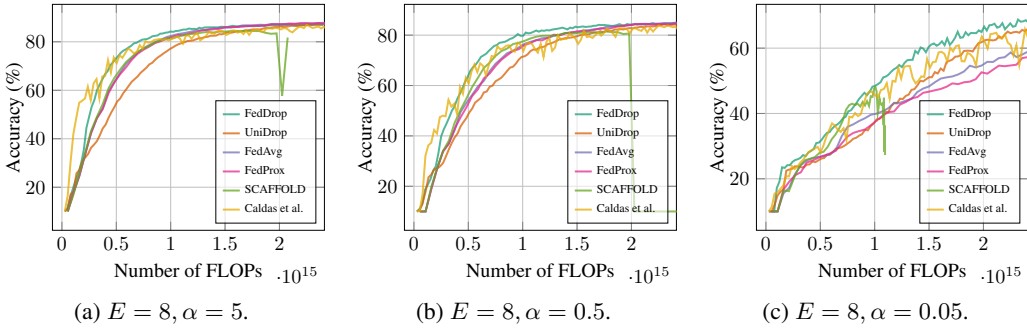

(a) $E = 8, \alpha = 5$.  (b) $E = 8, \alpha = 0.5$.  (c) $E = 8, \alpha = 0.05$.

Figure 6: Varying $\alpha$ for the dataset splits $\mathcal{D}_{|\mathbb{C}|}(\alpha)$ of CIFAR-10 with $E = 8$.

**Ablation Study.** To verify our design choice, we conduct a thorough set of ablations in Figure 7. We observe a certain degree of accuracy drop as these components are ablated, confirming the significance

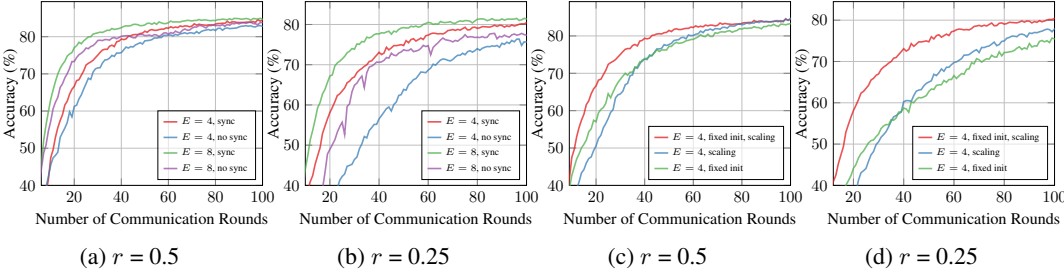

Figure 7: Ablation of design choices. (a, b) SyncDrop with and without synchronization. (c, d) with and without dropout scaling and fixed initialization under different FLOPs ratio $r$ for CIFAR-10.

of synchronization. This then further justifies the effectiveness of dropout scaling and initialization that comes with theoretical guarantees.

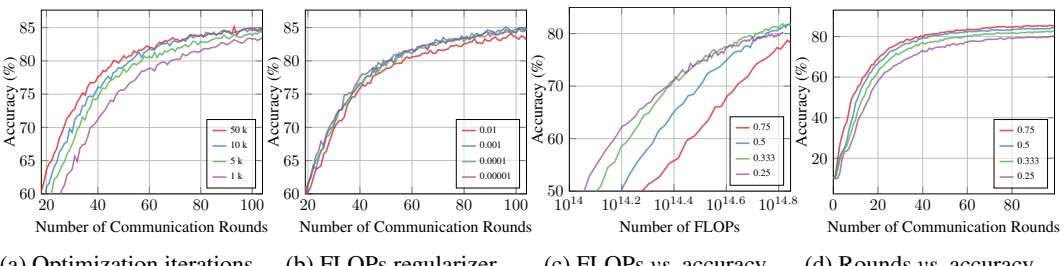

(a) Optimization iterations.   (b) FLOPs regularizer.   (c) FLOPs *vs.* accuracy.   (d) Rounds *vs.* accuracy.

Figure 8: Sensitivity analyses of (a) optimization iterations $I$, (b) FLOPs constraint regularizer $\mu$, and (c, d) the FLOPs budget ratio $r$.

**Sensitivity Analysis.** Figure 8 produces a sensitivity analysis of the hyperparameters introduced by our server-side probability objective minimization (7), namely the number of gradient descent iterations $I$ and the strength of regularization $\mu$. It turns out that additional iterations can improve convergence, which is desirable as the server compute budget is usually not the performance bottleneck. Moreover, a broad range of regularization $\mu$ do not have an adverse impact on accuracy, eliminating the need for a search. We also present an analysis on the effect of the FLOPs ratio $r$ as defined in (5). Reducing this ratio can sparsify models, for a faster initial convergence *w.r.t.* FLOPs. However, it comes at a cost of a slower convergence in latter rounds.

**Additional Results.** In Appendix F, we provide additional results and analyses. Specifically, Appendix F.1 includes the trade-off curves for all datasets under different accuracy budgets. We examine the performance impact on FL algorithms with different degrees of data imbalance (Appendix F.2), and varying hyperparameters such as local epochs per round $E$ (Appendix F.3) and the FLOPs budget ratio $r$ (Appendix F.4). Finally, Appendix F.5 provides full results with fractional device participation.

## 5 CONCLUSION

Recent federate learning (FL) algorithms focus on the reduction of communication costs, while neglecting the expensive local compute required by edge clients. This could be further exacerbated as the models we employ may increase in size over time to attain a higher task performance. We presented FedDrop, a novel FL technique that uses synchronous dropout with adaptive keep probabilities to concentrate the clients' training effort on neurons that they specialize well, while making the models sparse to reduce computational costs. Our experiments show that FedDrop can push the Pareto frontiers of communication/computation trade-off of FL scenarios notably further than existing algorithms. We believe this paves the way for future work that allow FL algorithms to scale up models being trained considerably, and consequently improve their performance on more challenging training tasks.

## 6 ETHICS STATEMENT

We believe FedDrop can have a positive impact on the environment as it aims to make federated learning more efficient.

The federated learning algorithms require communicating locally optimized client models to global server. In such a scenario, training data could potentially be recovered from model updates (Yin et al., 2021; Wang et al., 2020a), and this could have privacy implications. Although our work do not empirically address the privacy concern, we believe the added dropout layers do not bring noticeable impact to the underlying FL algorithm, for the reasons below:

- The dropout probabilities are computed on the server with client updates, the optimized probabilities are simply derived from information that are already collected by the base algorithm.
- There is no inter-client information exchange or leakage, as each client only receives its corresponding dropout probabilities.

## 7 REPRODUCIBILITY STATEMENT

We provide detailed information about model configuration in Appendix E, dataset splitting and partitioning in Section 4, and hyperparameters for optimization in Section 4. We try to justify the design choices of our method with ablation in Section 4, and all the experiments reported in this manuscript are conducted exhaustively with broad choices of hyperparameters to ensure fair evaluation for all methods. The code would be available during the review period and made public upon acceptance.

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

## A  TABLE OF NOTATIONS

For your convenience, we provide a list of notations used in this paper in Table 3.

Table 3: Summary of notations.

| Symbol | Description |
|---|---|
| $c, i, j$ | an individual client. |
| $\mathbb{C}$ | set of clients. |
| $\mathbf{C}$ | set of participating clients. |
| $\ell_c(\cdot)$ | training loss function of a client $c$. |
| $B$ | mini-batch size. |
| $\mathbb{D}_c$ | the local training dataset of client $c$. |
| $\boldsymbol{\theta}_c$ | the parameters of all layers in the model of client $c$. |
| $\mathsf{L}$ | softmax-cross-entropy (SCE) loss function. |
| $f$ | the feed-forward model. |
| $f^{[l]}$ | $l^{\text{th}}$ layer of model $f$. |
| $L$ | the total number of layers of model $f$. |
| $\mathbf{x}, \mathbf{x}^{[0]}$ | the model input. |
| $\mathbf{x}^{[L]}$ | the model output. |
| $\boldsymbol{\theta}_c^{(r+1)}$ | the parameters of client $c$ in round $r + 1$. |
| $\eta$ | learning rate. |
| $E$ | epochs of local training. |
| $r$ | the FLOPs budget ratio as defined in Section 3.4, $(0, 1]$. |
| $\phi$ | the fraction of participating clients, $(0, 1]$. |
| $\lambda_c$ | weight of client c proportional to the size of its training data set $|\mathbb{D}_c|$. |
| $\mathbf{p}, \mathbf{q}$ | (matrix) dropout keep probabilities. |
| $\mathbf{p}_c$ | (vector) dropout keep probabilities of client $c$. |
| $\mathbf{p}_c^{[l]}$ | (vector) a slice of $\mathbf{p}_c$ indicating the probabilities of all channels in the $l^{\text{th}}$ layer. |
| $\mathbf{p}_c^n$ | (scalar) the dropout keep probability of neuron $n$ in client $c$. |
| $\mathbf{t}^{[l]}$ | thresholding vector for $l^{\text{th}}$ layer with $n$ neurons. |
| $\mathbb{1}[\mathbf{z}]$ | the (element-wise) indicator function. |
| $\mathbf{z}^{\circ-1}$ | the element-wise inverse of $\mathbf{z}$. |
| $\mathbf{d}_c^{[l]}$ | binary dropout decision vector of $l^{\text{th}}$ layer. |
| $\hat{f}$ | sparsified model. |
| $g(\cdot)$ | FLOPs-budget constraint function. |
| $\hat{\ell}_c$ | loss function of client $c$. |
| $\mathbf{S}^{(r+1)}$ | (tensor) trajectory similarity with shape $|\mathbb{N}| \times |\mathbf{C}| \times |\mathbf{C}|$ as in Figure 2(b) for the $(r + 1)^{\text{th}}$ round. |
| $\mathbf{S}_{ij}^{n,(r+1)}$ | (scalar) trajectory similarity of neuron $n$ between pairs of clients $(i, j)$ in the $(r + 1)^{\text{th}}$ round. |
| $\hat{\mathbf{S}}$ | (tensor) estimated trajectory similarity accounting for the dropout variances. |
| $\boldsymbol{\vartheta}^{[l]}$ | flattened array of weight parameters of $l^{\text{th}}$ layer. |
| $\mathcal{D}_{|\mathbb{C}|}(\alpha)$ | Dirichlet distribution parameterized by $\alpha$ and $|\mathbb{C}|$. |
| $\mathbf{T}^n$ | a positive semidefinite matrix. |
| $\mathbf{\Pi}$ | a permutation matrix. |
| $\mathbf{h}^k$ | boolean vector. |
| $\mathbf{H}(\cdot)$ | the Hessian matrix *w.r.t.* parameters. |

## B  COMPUTING THE NUMBER OF FLOPS

In a sparse layer $\hat{f}^{[l]}$, the expected number of FLOPs per image-step is the sum of the flops required by both the convolution and ReLU activation:

$$\text{flops}(\hat{f}^{[l]}) = 2(\hat{C}^{[l]}\hat{C}^{[l-1]}K^{[l]^2}H^{[l]}W^{[l]} + \hat{C}^{[l]}H^{[l]}W^{[l]}), \tag{9}$$

where $K^2$ is the 2-dimensional kernel size, and $H \times W$ is the output feature map size. Moreover, $\hat{C}^{[l-1]}$ and $\hat{C}^{[l]}$ are the number of active input and output channels respectively; we expect $\hat{C}^{[l]}$, the average number of remaining output channels for layer $l$, to be $C^{[l]} \, \mathsf{mean}(\mathbf{p}_c^{[l]})$, where $\mathsf{mean}(\mathbf{z})$ takes the mean of elements in $\mathbf{z}$ and $C^{[l]}$ is the number of total output channels of layer $l$.

To generalize, we can rewrite the total number of FLOPs required by the overall model $\hat{f}$ to be:

$$\text{flops}(\hat{f}, \mathbf{p}_c) = 2\sum_{l=0}^{L} C^{[l]} \, \mathsf{mean}(\mathbf{p}_c^{[l]}) \left( K^{[l]^2} C^{[l-1]} \, \mathsf{mean}(\mathbf{p}_c^{[l-1]}) + 1 \right) H^{[l]} W^{[l]}, \quad (10)$$

and assume $\mathbf{p}^{[0]} = \mathbf{1}$ and $\mathbf{p}^{[L]} = \mathbf{1}$, since both the input and output of the model are considered to be dense. Finally, by setting $\mathbf{p} = \mathbf{1}$ for the entire model, we can get the number of FLOPs consumed by the overall model per image.

## C  PROOFS

### C.1  MODEL PARAMETER INITIALIZATION

Here, we restate Theorem 1 from Section 3.5:

**Theorem 1** (Parameter initialization). *To avoid vanishing/exploding gradients at initialization, the variance of the parameters of the $l^{th}$ layer followed by a dropout layer with keep probability $p$ should be $2p/N$, where $N = C^{[l-1]} K^{[l-1]^2}$ is the size of the receptive field of each output channel.*

The theorem proof depends on the following definitions and assumptions.

**Definition 1.** *A convolutional layer with channel dropout decisions $\mathbf{d}^{[l]}$ can be written as:*

$$\mathbf{x}^{[l+1]} \triangleq \mathsf{relu}(\mathbf{z}^{[l]}), \quad \text{where } \mathbf{z}^{[l]} \triangleq \mathbf{p}_c^{[l]\circ-1} \circ \mathbf{d} \circ (\boldsymbol{\vartheta}^{[l]} \mathbf{x}^{[l]} + \mathbf{b}^{[l]}), \quad (11)$$

*where we flatten the input features and the convolutional weights to be respectively $\mathbf{z}^{[l]}, \mathbf{x}^{[l]} \in \mathbb{R}^{N \times 1}$ and $\boldsymbol{\vartheta}^{[l]} \in \mathbb{R}^{C^{[l+1]} \times N^{[l]}}$, with $N^{[l]} = K^{[l]^2} C^{[l]}$ being the size of the input receptive field. Recall that $\circ$ indicates the element-wise product, and the notation $\mathbf{z}^{\circ-1}$ is the element-wise inverse of $\mathbf{z}$. Here each dropout decision in $\mathbf{d}^{[l]}$ is sampled with:*

$$\mathbf{d}^{[l]} \triangleq \mathbb{1}[\mathbf{t}^{[l]} < \mathbf{p}^{[l]}]. \quad (12)$$

At initialization, we assume that $\mathbf{p}^{[l]}$ shares a constant dropout keep probability $p$, and we can immediately derive that for any channel neuron $i$ at initialization:

$$\mathsf{E}[\mathbf{d}_i^{[l]}] = p \cdot 1 + (1 - p) \cdot 0 = p, \quad (13)$$

$$\mathsf{var}(\mathbf{d}_i^{[l]}) = \mathsf{E}[\mathbf{d}_i^{[l]^2}] - \mathsf{E}[\mathbf{d}_i^{[l]}]^2 = p \cdot 1 + (1 - p) \cdot 0 - p^2 = p - p^2. \quad (14)$$

**Assumption 1.** *We assume the model input $\mathbf{x}^{[0]}$ and for each layer $l$ the intermediate values $\mathbf{z}^{[l]}$ to be IID and all have zero-valued means. We also let the individual weight parameters $\boldsymbol{\vartheta}^{[l]}$ to be initialized with IID and zero means, and the biases $\mathbf{b}^{[l]} = \mathbf{0}$. Therefore, for each layer $l \in [1, L]$:*

$$\mathsf{E}(\mathbf{z}_{ij}^{[l]}) = 0 \quad \text{for } \mathbf{z}_{ij}^{[l]} \in \mathbf{z}^{[l]}, \quad (15)$$

$$\mathsf{E}(\boldsymbol{\vartheta}_{ij}^{[l]}) = 0 \quad \text{for } \boldsymbol{\vartheta}_{ij}^{[l]} \in \boldsymbol{\vartheta}^{[l]}. \quad (16)$$

**Assumption 2.** *To avoid vanishing/exploding gradients, we restrict the variances of each value in $\mathbf{z}^{[l]}$ to be constant across all layers $l \in [1, L]$, i.e.*

$$\mathsf{var}(\mathbf{z}_i^{[m]}) = \mathsf{var}(\mathbf{z}_j^{[n]}) \quad \text{for } m, n \in [1, L], \mathbf{z}_i^{[m]} \in \mathbf{z}^{[m]}, \mathbf{z}_j^{[n]} \in \mathbf{z}^{[n]}. \quad (17)$$

We adapt the proofs from (He et al., 2015; Glorot & Bengio, 2010) by taking into consideration the variance introduced by dropout layers:

*Proof.* By Definition 1, for any value $\mathbf{z}_i^{[l]} \in \mathbf{z}^{[l]}$, at initialization:

$$\text{var}(\mathbf{z}_i^{[l]}) = \tfrac{1}{p^2}\,\text{var}(\textstyle\sum_j \mathbf{d}_i^{[l]}\boldsymbol{\vartheta}_{ij}^{[l]}\mathbf{x}_j^{[l]} + \mathbf{d}_i^{[l]}\mathbf{b}_i^{[l]})$$

As $\mathbf{d}_i^{[l]}$, $\boldsymbol{\vartheta}_{ij}^{[l]}$ and $\mathbf{x}_j^{[l]}$ are independent, and $\mathbf{b}^{[l]} = \mathbf{0}$ at initialization following Assumption 1:

$$= \tfrac{N^{[l]}}{p^2}\,\text{var}(\mathbf{d}_i^{[l]}\boldsymbol{\vartheta}_{ij}^{[l]}\mathbf{x}_j^{[l]})$$

$$= \tfrac{N^{[l]}}{p^2}\big(\text{var}(\mathbf{d}_i^{[l]}) + \mathsf{E}[\mathbf{d}_i^{[l]}]^2\big)\big(\text{var}(\boldsymbol{\vartheta}_{ij}^{[l]}) + \mathsf{E}[\boldsymbol{\vartheta}_{ij}^{[l]}]^2\big)\big(\text{var}(\mathbf{x}_j^{[l]}) + \mathsf{E}[\mathbf{x}_j^{[l]}]^2\big)$$

$$- \mathsf{E}[\mathbf{d}_i^{[l]}]^2\,\mathsf{E}[\boldsymbol{\vartheta}_{ij}^{[l]}]^2\,\mathsf{E}[\mathbf{x}_j^{[l]}]^2. \tag{18}$$

Again following Assumption 1, $\mathsf{E}[\boldsymbol{\vartheta}_{ij}^{[l]}] = 0$. Also, $\text{var}(\mathbf{x}_j^{[l]}) + \mathsf{E}[\mathbf{x}_j^{[l]}]^2 = \mathsf{E}\big(\mathbf{x}_j^{[l]\,2}\big)$, $\mathsf{E}[\mathbf{d}_i^{[l]}] = p$, and $\text{var}(\mathbf{d}_i^{[l]}) = p(1-p)$:

$$\text{var}(\mathbf{z}_i^{[l]}) = \frac{N^{[l]}}{p}\,\text{var}(\boldsymbol{\vartheta}_{ij}^{[l]})\,\mathsf{E}\big(\mathbf{x}_j^{[l]\,2}\big). \tag{19}$$

As proven by (He et al., 2015) for ReLU-nonlinearity, $\mathsf{E}\big(\mathbf{x}_j^{[l]\,2}\big) = \tfrac{1}{2}\,\text{var}(\mathbf{z}_j^{[l-1]})$, and using Assumption 2, we can let $\text{var}(\mathbf{z}_i^{[l]}) = \text{var}(\mathbf{z}_j^{[l-1]})$, and solve (19) to give $\text{var}(\boldsymbol{\vartheta}_{ij}^{[l]}) = \tfrac{2p}{N^{[l]}}$. Finally, if we are using Gaussian distributions to initialize model parameters, then $\boldsymbol{\vartheta}_{ij}^{[l]} \sim \mathcal{N}\big(0, \sqrt{2p/N^{[l]}}\big)$. $\qquad\square$

## C.2 OPTIMIZATION OBJECTIVE

This section provides proof sketches for the optimization objective (7) described in Section 3.6. Let's assume $\boldsymbol{g}_c \triangleq \lambda_c\nabla_{\boldsymbol{\theta}}\ell_c$, and $\hat{\boldsymbol{g}}_c \triangleq \lambda_c\nabla_{\boldsymbol{\theta}}\hat{\ell}_c$ to simplify the derivations below.

**Lemma 1.** *The sample gradient with dropout $\hat{\boldsymbol{g}}_c$ can be approximated by $\mathbf{d}_c \circ \mathbf{p}_c^{\circ-1} \circ \boldsymbol{g}_c$.*

*Proof.* Consider the $1^{\text{st}}$-order Taylor expansion of $\hat{\ell}_c(\boldsymbol{\theta})$:

$$\hat{\ell}_c(\boldsymbol{\theta}_c) = \ell_c(\mathbf{d}_c \circ \mathbf{p}_c^{\circ-1} \circ \boldsymbol{\theta}_c) \approx \ell_c(\boldsymbol{\theta}_c) + \big(\nabla_{\boldsymbol{\theta}_c}\ell_c\big)^{\mathsf{T}}\big(\mathbf{d}_c \circ \mathbf{p}_c^{\circ-1} \circ \boldsymbol{\theta}_c - \boldsymbol{\theta}_c\big).$$

Differentiating both sides by $\boldsymbol{\theta}_c$, we have the following, where $\mathbf{H}\big(\ell_c\big)$ signify the Hessian of $\ell_c(\boldsymbol{\theta}_c)$:

$$\nabla_{\boldsymbol{\theta}_c}\hat{\ell}_c - \nabla_{\boldsymbol{\theta}_c}\ell_c \approx \nabla_{\boldsymbol{\theta}_c}\Big(\big(\nabla_{\boldsymbol{\theta}_c}\ell_c\big)^{\mathsf{T}}\big(\mathbf{d}_c \circ \mathbf{p}_c^{\circ-1} - \mathbf{1}\big) \circ \boldsymbol{\theta}_c\Big)$$

$$= \big(\nabla_{\boldsymbol{\theta}_c}\ell_c\big) \circ \big(\mathbf{d}_c \circ \mathbf{p}_c^{\circ-1} - \mathbf{1}\big) + \big((\mathbf{d}_c \circ \mathbf{p}_c^{\circ-1} - \mathbf{1}) \circ \boldsymbol{\theta}_c\big)^{\mathsf{T}}\mathbf{H}(\ell_c).$$

We omit the second term $\big((\mathbf{d}_c \circ \mathbf{p}_c^{\circ-1} - \mathbf{1}) \circ \boldsymbol{\theta}_c\big)^{\mathsf{T}}\mathbf{H}(\ell_c)$ for simplicity, as the Hessian of $\ell_c(\boldsymbol{\theta}_c)$ is compute-intensive, thus:

$$\hat{\boldsymbol{g}}_c - \boldsymbol{g}_c = \lambda_c\big(\nabla_{\boldsymbol{\theta}_c}\hat{\ell}_c - \nabla_{\boldsymbol{\theta}_c}\ell_c\big) \approx \boldsymbol{g}_c \circ \big(\mathbf{d}_c \circ \mathbf{p}_c^{\circ-1} - \mathbf{1}\big). \tag{20}$$

$\qquad\square$

In practice, we cannot obtain $\mathbf{S}_{ij}^n \triangleq \mathsf{E}\big[g_i^n g_j^n\big]$, as evaluating the gradients of dense models defeats the intention of accelerated training. We can instead estimate the value with the following lemma:

**Lemma 2.** $\mathsf{E}\big[g_i^n g_j^n\big]$ *can be approximated with* $\mathsf{E}\big[\hat{g}_i^n \hat{g}_j^n\big]\max(\mathbf{p}_i^n, \mathbf{p}_i^n)$, *where* $\mathbf{p}_i^n, \mathbf{p}_i^n$ *are the dropout keep probabilities from the previous round of the neuron $n$ of respective clients $i, j$.*

*Proof.* From Lemma 1, by rearranging (20), we have $\boldsymbol{g}_c \approx \hat{\boldsymbol{g}}_c \circ \mathbf{d}_c^{\circ-1} \circ \mathbf{p}_c$ for client $c$. Therefore,

$$\mathsf{E}\big[\hat{g}_i^n \hat{g}_j^n\big] \approx \mathsf{E}\big[\tfrac{g_i^n g_j^n \mathbf{d}_i^n \mathbf{d}_j^n}{\mathbf{p}_i^n \mathbf{p}_j^n}\big]$$

As $\mathbf{d}_c$ and $\boldsymbol{g}_c$ are independent for each client $c$,

$$= \mathsf{E}\big[g_i^n g_j^n\big]\,\mathsf{E}\big[\tfrac{\mathbf{d}_i^n \mathbf{d}_j^n}{\mathbf{p}_i^n \mathbf{p}_j^n}\big]$$

By the definition of the SyncDrop layer in Section 3.3,

$$= \mathsf{E}\big[g_i^n g_j^n\big] \int_0^1 \tfrac{\mathbf{1}[t \le \mathbf{p}_i^n]\,\mathbf{1}[t \le \mathbf{p}_j^n]}{\mathbf{p}_i^n \mathbf{p}_j^n}\,\mathrm{d}t$$

$$= \mathsf{E}\big[g_i^n g_j^n\big]\,\tfrac{\min(\mathbf{p}_i^n, \mathbf{p}_j^n)}{\mathbf{p}_i^n \mathbf{p}_j^n}.$$

Rearrange to give:

$$\mathsf{E}\big[g_i^n g_j^n\big] = \mathsf{E}\big[\hat{g}_i^n \hat{g}_j^n\big]\max(\mathbf{p}_i^n, \mathbf{p}_j^n). \tag{21}$$

$\qquad\square$

Finally, we reiterate Theorem 2 that introduces the approximate objective (7):

**Theorem 2** (Approximate Objective)**.** *Assuming that for a channel neuron $n \in \mathbb{N}$ and a pair of clients $i, j \in \mathbb{C}$, we have $\hat{\mathbf{S}}_{ij}^n \triangleq \lambda_i \lambda_j \max(\mathbf{p}_i^n, \mathbf{p}_j^n) \Delta \boldsymbol{\theta}_i^n \cdot \Delta \boldsymbol{\theta}_j^n$, where $\Delta \boldsymbol{\theta}_i^n, \Delta \boldsymbol{\theta}_j^n$ are the parameter updates after a round of FL training, $\mathbf{p}_i^n, \mathbf{p}_j^n$ are the current dropout keep probabilities, and $\max(\mathbf{x}, \mathbf{y})$ represents the element-wise max between $\mathbf{x}$ and $\mathbf{y}$, the optimization objective of (6) can be approximated by:*

$$\min_{\mathbf{q}} \mathsf{obj}(\hat{\mathbf{S}}, \mathbf{q}) \triangleq \min_{\mathbf{q}} \sum_{n \in \mathbb{N}, i,j \in \mathbb{C}} \hat{\mathbf{S}}_{ij}^n / \max(\mathbf{q}_i^n, \mathbf{q}_j^n). \tag{7}$$

*Proof.* We start by deriving from (6):

$$\min_{\mathbf{p}} \mathsf{E} \left\| \sum_{c \in \mathbb{C}} \lambda_c \nabla_{\boldsymbol{\theta}_c} \hat{\ell}_c - \sum_{c \in \mathbb{C}} \lambda_c \nabla_{\boldsymbol{\theta}_c} \ell_c \right\|_2^2$$

$$= \mathsf{E} \left[ \left( \sum_{c \in \mathbb{C}} (\hat{\boldsymbol{g}}_c - \boldsymbol{g}_c) \right)^{\mathsf{T}} \left( \sum_{c \in \mathbb{C}} (\hat{\boldsymbol{g}}_c - \boldsymbol{g}_c) \right) \right]$$

$$= \mathsf{E} \left[ \sum_{i,j \in \mathbb{C}} (\hat{\boldsymbol{g}}_i - \boldsymbol{g}_i)^{\mathsf{T}} (\hat{\boldsymbol{g}}_j - \boldsymbol{g}_j) \right],$$

following Lemma 1,

$$\approx \sum_{i,j \in \mathbb{C}} \mathsf{E} \left[ \left( \boldsymbol{g}_i \circ \left( \mathbf{d}_i \circ \mathbf{q}_i^{\circ -1} - \mathbf{1} \right) \right)^{\mathsf{T}} \left( \boldsymbol{g}_j \circ \left( \mathbf{d}_j \circ \mathbf{q}_j^{\circ -1} - \mathbf{1} \right) \right) \right]$$

$$= \sum_{i,j \in \mathbb{C}, n \in \mathbb{N}} \mathsf{E} \left[ \boldsymbol{g}_i^n \boldsymbol{g}_j^n \left( \frac{\mathbf{d}_i^n}{\mathbf{q}_i^n} - 1 \right) \left( \frac{\mathbf{d}_j^n}{\mathbf{q}_j^n} - 1 \right) \right],$$

as $\mathbf{d}_c$ and $\boldsymbol{g}_c$ are independent for each client $c$, and following Lemma 2,

$$= \sum_{i,j \in \mathbb{C}, n \in \mathbb{N}} \mathsf{E} \left[ \hat{\boldsymbol{g}}_i^n \hat{\boldsymbol{g}}_j^n \right] \max(\mathbf{p}_i^n, \mathbf{p}_j^n) \mathsf{E} \left[ \left( \frac{\mathbf{d}_i^n}{\mathbf{q}_i^n} - 1 \right) \left( \frac{\mathbf{d}_j^n}{\mathbf{q}_j^n} - 1 \right) \right],$$

by definition of the SyncDrop layer in Section 3.3,

$$= \sum_{i,j \in \mathbb{C}, n \in \mathbb{N}} \mathsf{E} \left[ \hat{\boldsymbol{g}}_i^n \hat{\boldsymbol{g}}_j^n \right] \max(\mathbf{p}_i^n, \mathbf{p}_j^n) \int_0^1 \left( \frac{\mathbf{1}[t \leq \mathbf{q}_i^n]}{\mathbf{q}_i^n} - 1 \right) \left( \frac{\mathbf{1}[t \leq \mathbf{q}_j^n]}{\mathbf{q}_j^n} - 1 \right) \mathrm{d}t$$

$$= \sum_{i,j \in \mathbb{C}, n \in \mathbb{N}} \mathsf{E} \left[ \hat{\boldsymbol{g}}_i^n \hat{\boldsymbol{g}}_j^n \right] \max(\mathbf{p}_i^n, \mathbf{p}_j^n) \left( \frac{1}{\max(\mathbf{q}_i^n, \mathbf{q}_j^n)} - 1 \right) \tag{22}$$

Finally, as it is impractical to use single step gradients $\hat{\boldsymbol{g}}_i^n$ and $\hat{\boldsymbol{g}}_j^n$ on the server, we approximate them with the respective parameter updates after one round of training, *i.e.* $\lambda_i \Delta \boldsymbol{\theta}_i^n$ and $\lambda_j \Delta \boldsymbol{\theta}_j^n$; recall the definition of $\hat{\mathbf{S}}$, we have

$$\approx \sum_{i,j \in \mathbb{C}, n \in \mathbb{N}} \frac{\hat{\mathbf{S}}_{ij}^n}{\max(\mathbf{q}_i^n, \mathbf{q}_j^n)} - \sum_{i,j \in \mathbb{C}, n \in \mathbb{N}} \mathbf{S}_{ij}^n. \tag{23}$$

The first term is exactly (7), and the second term is a constant and can be safely removed from the optimization objective. $\square$

The value $\hat{\mathbf{S}}_{ij}^n$ can be considered as the similarity of the model updates between clients $i$ and $j$ for the group neuron $n$. Intuitively, from the perspective of an individual neuron $n$ and the two clients $i$ and $j$, minimizing the term $\hat{\mathbf{S}}_{ij}^n / \max(\mathbf{p}_i^n, \mathbf{p}_j^n)$ incentivizes $\mathbf{p}_i^n$ and $\mathbf{p}_j^n$ to grow when $\hat{\mathbf{S}}_{ij}^n$ is positive, and to become smaller when it is negative.

With the derivations above, we can proceed to prove the objective in Theorem 2 is bounded with a minimum. First, we can show that the matrix $\mathbf{S}^n$ is positive semidefinite for each neuron $n \in \mathbb{N}$.

**Lemma 3.** *The matrix $\mathbf{S}^n$ is positive semidefinite for all $n \in \mathbb{N}$.*

*Proof.* By definition, $\mathbf{S}^n = \mathsf{E}[\boldsymbol{g}^n \boldsymbol{g}^{n\mathsf{T}}]$, where $\boldsymbol{g}^n$ comprises the gradient vectors of the $n^{\text{th}}$ neuron across clients $c \in \mathbb{C}$. The matrix $\mathbf{S}^n$ is therefore positive semidefinite, as it is an averaged sum of samples of positive semidefinite matrices $\boldsymbol{g}^n \boldsymbol{g}^{n\mathsf{T}}$. $\square$

**Theorem 3** (Optimization bound)**.** *The objective $\min_{\mathbf{q}} \sum_{nij} \hat{\mathbf{S}}_{ij}^n / \max(\mathbf{q}_i^n, \mathbf{q}_j^n)$ is trivially minimized to $\sum_{nij} \hat{\mathbf{S}}_{ij}^n$ when $\mathbf{q}_c^n = 1$ for $\forall n \in \mathbb{N}, c \in \mathbf{C}$.*

*Proof.* Without loss of generality, we can permute the matrix $\hat{\mathbf{S}}^n$ and the probability vector $\mathbf{p}^n$ to assume a descending order on the probabilities. Let $\mathbf{T}^n \triangleq \mathbf{\Pi}\hat{\mathbf{S}}^n\mathbf{\Pi}^\mathsf{T}$ and $\boldsymbol{\pi}^n \triangleq \mathbf{\Pi}\mathbf{p}^n$ denote the respective permuted trajectory similarity and probability variants, where $\mathbf{\Pi} \in \{0,1\}^{|\mathbb{C}|\times|\mathbb{C}|}$ is the permutation matrix, such that

$$\sum_{nij} \frac{\hat{\mathbf{S}}^n_{ij}}{\max(\mathbf{p}^n_i,\mathbf{p}^n_j)} = \sum_{nij} \frac{\mathbf{T}^n_{ij}}{\max(\boldsymbol{\pi}^n_i,\boldsymbol{\pi}^n_j)} \quad \text{and} \quad \boldsymbol{\pi}^n_1 \geq \boldsymbol{\pi}^n_2 \geq \cdots \geq \boldsymbol{\pi}^n_{|\mathbb{C}|}. \tag{24}$$

We introduce a boolean vector $\mathbf{h}^k \in \mathbb{R}^{|\mathbb{C}|}$, where each element $\mathbf{h}^k_c$ of it is defined as:

$$\mathbf{h}^k_c \triangleq \mathbb{1}[\boldsymbol{\pi}^n_k \geq \boldsymbol{\pi}^n_c],$$

we then let $\mathbf{H}^k \triangleq \mathbf{h}^k\mathbf{h}^{k\mathsf{T}}$, and it follows that $\mathbf{H}^k_{ij} = \mathbf{h}^k_i\mathbf{h}^k_j = \mathbb{1}[\boldsymbol{\pi}^n_k \geq \boldsymbol{\pi}^n_i]\,\mathbb{1}[\boldsymbol{\pi}^n_k \geq \boldsymbol{\pi}^n_j]$. Hence, we can rewrite:

$$\begin{aligned}
\sum_{nij} \frac{\mathbf{T}^n_{ij}}{\max(\boldsymbol{\pi}^n_i,\boldsymbol{\pi}^n_j)} &= \sum_{nij} \mathbf{T}^n_{ij}\min\left(\frac{1}{\boldsymbol{\pi}^n_i},\frac{1}{\boldsymbol{\pi}^n_j}\right) \\
&= \sum_{nij} \mathbf{T}^n_{ij}\left(\sum_{k=1}^{|\mathbb{C}|-1} \frac{\mathbf{H}^k_{ij}-\mathbf{H}^{k+1}_{ij}}{\boldsymbol{\pi}^n_k} + \frac{\mathbf{H}^{|\mathbb{C}|}_{ij}}{\boldsymbol{\pi}^n_{|\mathbb{C}|}}\right) \\
&= \sum_{nij} \mathbf{T}^n_{ij}\left(\sum_{k=1}^{|\mathbb{C}|-1} \frac{\mathbf{H}^k_{ij}}{\boldsymbol{\pi}^n_k} - \sum_{k=1}^{|\mathbb{C}|-1} \frac{\mathbf{H}^{k+1}_{ij}}{\boldsymbol{\pi}^n_k} + \frac{\mathbf{H}^{|\mathbb{C}|}_{ij}}{\boldsymbol{\pi}^n_{|\mathbb{C}|}}\right) \\
&= \sum_{nij} \mathbf{T}^n_{ij}\left(\frac{\mathbf{H}^1_{ij}}{\boldsymbol{\pi}^n_1} + \sum_{k=2}^{|\mathbb{C}|} \mathbf{H}^k_{ij}\left(\frac{1}{\boldsymbol{\pi}^n_k} - \frac{1}{\boldsymbol{\pi}^n_{k-1}}\right)\right) \\
&= \sum_n \left(\sum_{ij} \frac{\mathbf{T}^n_{ij}}{\boldsymbol{\pi}^n_1} + \sum_{k=2}^{|\mathbb{C}|} \sum_{ij} \mathbf{T}^n_{ij}\mathbf{h}^k_i\mathbf{h}^k_j\left(\frac{1}{\boldsymbol{\pi}^n_k} - \frac{1}{\boldsymbol{\pi}^n_{k-1}}\right)\right) \\
&= \sum_n \left(\frac{\mathbf{1}^\mathsf{T}\mathbf{T}^n\mathbf{1}}{\boldsymbol{\pi}^n_1} + \sum_{k=2}^{|\mathbb{C}|} \mathbf{h}^{k\mathsf{T}}\mathbf{T}^n\mathbf{h}^k\left(\frac{1}{\boldsymbol{\pi}^n_k} - \frac{1}{\boldsymbol{\pi}^n_{k-1}}\right)\right). \tag{25}
\end{aligned}$$

It is notable that $\frac{1}{\boldsymbol{\pi}^n_c} \geq 1$ for all $c \in \mathbb{C}$. The first term is bounded by a non-negative $\mathbf{1}^\mathsf{T}\mathbf{T}^n\mathbf{1}$ *i.e.* the sum of all elements in a positive semidefinite matrix. The second term is also non-negative, as $\mathbf{h}^{k\mathsf{T}}\mathbf{T}^n\mathbf{h}^k \geq 0$ by the definition of a positive semidefinite matrix $\mathbf{T}^n$, and $\frac{1}{\boldsymbol{\pi}^n_k} - \frac{1}{\boldsymbol{\pi}^n_{k-1}} \geq 0$ by the ordering proposed in (24). Notably if $\boldsymbol{\pi}^n_c = 1$ for all $n \in \mathbb{N}$ and $c \in \mathbb{C}$, then the objective is trivially minimized, with a minimum value $\sum_{nij} \mathbf{T}^n_{ij} = \sum_{nij} \mathbf{S}^n_{ij}$. We therefore include the FLOPs budgets discussed in Section 3.4 and Appendix B into our optimizations to solve for non-trivial $\mathbf{p}$. $\qquad\square$

# D  THE FEDDROP ALGORITHM

## D.1  HIGH-LEVEL OVERVIEW

Algorithm 1 provides an overview of the FedDrop FL algorithm. It accepts the client loss functions $\hat{\ell}_c$ with SyncDrop layers, client data weights $\lambda_c$, the FLOPS constraints function $g : [0,1]^{\mathbb{C}\times\mathbb{N}} \to \mathbb{R}$, the SGD learning rate $\eta$, the number of local epochs $E$ the number of FL rounds $R$, and a client sub-sampling ratio $\phi$. It returns optimized model parameters $\boldsymbol{\theta}^{(R+1)}$ on the final round.

The algorithm starts by initializing model parameters $\boldsymbol{\theta}^{(0)}$ to be shared across all clients, and assigns a uniform keep probability to $\mathbf{p}$ for all SyncDrop layers, which satisfies the FLOPs constraint $g(\mathbf{p}) = 0$ (lines 2 and 3). After collecting trained model parameters $\boldsymbol{\theta}_c^{(r+1)}$ from the current round $r + 1$ (line 6), the FedDrop server computes $\Delta\boldsymbol{\theta}_c^{(r+1)}$, *i.e.* the difference between the averaged model from the previous round $\boldsymbol{\theta}^{(r)}$ with the trained model $\boldsymbol{\theta}_c^{(r+1)}$ for client $c$ (line 7, also in Figure 2a).

It then measures the trajectory similarity $\hat{\mathbf{S}}_{ij}^{n\,(r+1)}$ for each neuron $n$ between each pair of clients $(i,j)$, as described on line 11 and illustrated in Figure 2b. The term $\lambda_i\Delta\boldsymbol{\theta}_i^{n\,(r+1)} \cdot \lambda_j\Delta\boldsymbol{\theta}_j^{n\,(r+1)}$ computes the dot-product between the weighted parameter updates of clients $i$ and $j$, and $\max\left(\mathbf{p}_i^{n\,(r)}, \mathbf{p}_j^{n\,(r)}\right)$ adjusts for the covariance from the synchronized dropouts of the previous round. Finally, the objective $\arg\min_{\mathbf{q}} \mathrm{obj}(\hat{\mathbf{S}}, \mathbf{q})$ is optimized to minimize the impact of model sparsity on convergence.

## D.2  COMPUTATIONAL OVERHEAD OF THE OPTIMIZATION OBJECTIVE

The complexity of computing the trajectory similarity in each round is $O(W|\mathbf{C}|^2)$, where $W$ is the number of parameters and $|\mathbf{C}|$ is the number of participating clients. Additionally, the probability

---

**Algorithm 1** The FedDrop algorithm.

---

1: **function** FEDAVGDROP($\{(\hat{\ell}_c, \lambda_c) \colon c \in \mathbb{C}\}, g, \eta, E, R, \phi$)
2:      random_initialize($\boldsymbol{\theta}^{(0)}, \mathbf{p}$)          ▷ Initialize parameters and a uniform keep probability.
3:      **for** $r \leftarrow 1, 2, \ldots R$ **do**
4:          $\mathbf{C} \leftarrow$ subsample($\mathbb{C}, \phi$)          ▷ Sample a subset of clients.
5:          **par for** $c \in \mathbf{C}$ **do**
6:              $\boldsymbol{\theta}_c^{(r+1)} \leftarrow \mathsf{SGD}_c\big(\hat{\ell}_c, \boldsymbol{\theta}^{(r)}, \mathbf{p}_c, \eta, E\big)$      ▷ Client training with SyncDrop layers.
7:              $\Delta\boldsymbol{\theta}_c^{(r+1)} \leftarrow \boldsymbol{\theta}^{(r)} - \boldsymbol{\theta}_c^{(r+1)}$          ▷ Client update trajectories.
8:          **end par for**
9:          $\boldsymbol{\theta}^{(r+1)} \leftarrow \sum_{c \in \mathbf{C}} \lambda_c \boldsymbol{\theta}_c^{(r+1)}$          ▷ Server aggregation with FedAvg.
10:          **for** $i \in \mathbf{C}, j \in \mathbf{C}, n \in \mathbb{N}$ **do**          ▷ Trajectory-similarity estimation.
11:              $\hat{\mathbf{S}}_{ij}^n \leftarrow \lambda_i \lambda_j \max\big(\mathbf{p}_i^n, \mathbf{p}_j^n\big) \Delta\boldsymbol{\theta}_i^{u\,(r+1)} \cdot \Delta\boldsymbol{\theta}_j^{u\,(r+1)}$
12:          **end for**          ▷ Optimal dropout probabilities.
13:          $\mathbf{p} \leftarrow \mathrm{argmin}_{\mathbf{q}}\, \mathrm{obj}(\hat{\mathbf{S}}, \mathbf{q})$ *s.t.* $g(\mathbf{q}) \geq 0$      ▷ Objective from (7)
14:      **end for**
15:      **return** $\boldsymbol{\theta}^{(R+1)}$
16: **end function**

---

optimization stage on the server has a complexity of $O(I|\mathbb{N}||\mathbf{C}|^2)$, where $I$ is the number of optimization iterations, $|\mathbb{N}|$ is the number of channels in the model. As an example, the VGG-9 model with 20 clients and $I = 10k$ steps of optimization would require only 53 G FLOPs for computing the trajectory similarity, and $\leq 448$ G FLOPs for the optimization steps, which can often exit early because it has converged. In practice, the server optimization only takes less than 1 minute on average. For reference, if we use local epochs per round $E = 8$, and a FLOPs ratio $r = 0.5$, the 20 clients require a total of 25.8 T FLOPs per round. Overall, under this setting the additional computation overhead incurred by the server is only about 2% of total computational costs. For more practical consideration, it is noteworthy that typically, the server is much less constrained in terms of computational power compared to the edge clients. In addition, given the above example, a single client requires 1.29 T FLOPs and the server requires 448 G FLOPs in total. The former consumes multiple hours of training in a realistic implementation on an iPhone 12 Pro (extrapolating from the profiling result in Table 10), whereas the latter only requires minutes to complete on an Nvidia V100 GPU.

### D.3    AN EXAMPLE HISTORY OF PROBABILITY DISTRIBUTION

To better understand the probability optimization, we plot the histogram of optimized probabilities for each training round in Figure 9. It reveals that after multiple rounds of training, FedDrop gradually learns the importance of channels and start to make certain channels contribute less to the model training. Note that the histograms only provides distributions of probability values of all clients, each client may have vastly different optimized probabilities for the same channels.

### D.4    MEMORY REDUCTION

As SyncDrop layers can sparsify models, many channels in the activation map can simply be skipped and never stored in both the forward and backward passes. For this reason, FedDrop can further reduce the memory cost associated with training. As an example, VGG-9 equipped with SyncDrop can see up to 40% reduction in the memory requirement for training. We summarized the memory reduction for different scenarios in Table 4.

### D.5    ACCURACY OF FLOPS ESTIMATION

To confirm the accuracy of our FLOPs estimation with the expected values, in Table 5 we sample the dropout layers in the Fashion-MNIST model with FLOPs budget ratios $r \in \{0.25, 0.5, 0.75\}$, can compute the actual FLOPs counts. Under the setting of $R = 100$ training rounds and local epochs

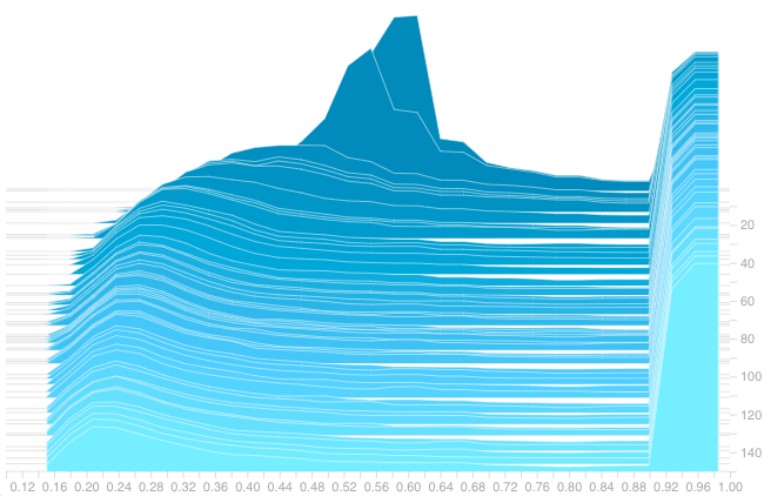

Figure 9: We present the evolution of dropout keep probabilities (horizontal axis) for multiple rounds of training (vertical axis) for VGG-9 training with FLOPS budget ratio $r = 0.5$ and the number of local epochs per round $E$ is $4$. After a few rounds of training, FedDrop can learn to distinguish channel importance.

Table 4: Theoretical memory reductions of FedDrop with varying batch sizes.

| Batch size | Feature (MB) | Params (MB) | Total (MB) | FLOPs ratio | FedDrop Total (MB) | Reduction (%) |
|---|---|---|---|---|---|---|
| 4 | 3.02 | 5.82 | 8.83 | 0.75 | 8.08 | 8.53 |
| | | | | 0.5 | 7.33 | 17.07 |
| 8 | 6.03 | 5.82 | 11.85 | 0.75 | 10.34 | 12.72 |
| | | | | 0.5 | 8.83 | 25.45 |
| 16 | 12.06 | 5.82 | 17.88 | 0.75 | 14.87 | 16.86 |
| | | | | 0.5 | 11.85 | 33.73 |
| 32 | 24.13 | 5.82 | 29.95 | 0.75 | 23.91 | 20.14 |
| | | | | 0.5 | 17.88 | 40.28 |

per round $E = 8$. We found the error to be within $0.06\%$. The low deviation of the expected value from the actual result is an effect of the law of large numbers.

Table 5: Comparing the estimated and actual FLOPs counts of FedDrop. The figures within parentheses represent the sampled FLOPs of 3 experiments with different random seeds.

| FLOPs budget ratio $r$ | 0.25 | 0.5 | 0.75 |
|---|---|---|---|
| Expected (T FLOPs) | 140.97 | 281.85 | 422.50 |
| Sampled (T FLOPs) | (140.88, 140.91, 140.92 ) | (281.78, 281.81, 281.78) | (422.47, 422.48, 422.46) |
| Relative error (mean) | 0.047% | 0.021% | 0.0071% |
| Relative error (std) | 0.012% | 0.005% | 0.0019% |

## E    EXPERIMENT CONFIGURATIONS

**Models.** Here, we provide the layout of the models used by the respective datasets. All models listed for computer vision (CV) tasks (in Tables 6 to 8) are feed-forward CNNs. Table 9 shows the character-level single-layer LSTM model used for the natural language processing (NLP) task. Note that sparsity for the LSTM model were introduced for the hidden state and memory cell neurons.

**Datasets.** The experiments are carried out on three popular CV datasets (CIFAR-10 (Krizhevsky et al., 2014), Fashion-MNIST (Xiao et al., 2017) and SVHN (Netzer et al., 2011)), and a natural language dataset (Shakespeare (Caldas et al., 2018b)). We use the standard train/test split on these datasets and report accuracy on test dataset with a global model on server-side. To simulate different heterogeneity levels, for CV tasks we split the datasets by class label with the distribution $\mathcal{D}_{|\mathbb{C}|}(\alpha)$ (Hsu et al., 2019; Wang et al., 2020b), and adjust the $\alpha$ constant for varying the degree of data heterogeneity, $\alpha = 0.5$ unless specified for all experiments. Simple data augmentation was also employed for the training images, including random crop and normalization, and random flip in addition for CIFAR-10. For the Shakespeare NLP dataset, we follow LEAF (Caldas et al., 2018b) for next-character prediction, and consider each speaking role in each play as an individual client, and the resulting splits are thus inherently heterogeneous since each local client only contains the corresponding role's sentences.

**Hyperparameters.** We consider different hyperparameter settings for different tasks:

- *Client Sampling.* For full client participation, we use 20 clients for CIFAR-10 and 100 clients for the remaining CV and NLP tasks. To simulate the fractional client participation, we randomly sample 10 out of 1000 clients on all datasets for CV tasks and 10 out of 660 clients on Shakespeare dataset for NLP task.
- *Optimization.* As for CV tasks, we searched the hyperparameters and found setting batch size $B = 4$ and learning rate $\eta = 0.02$ were universally optimal for most methods. For NLP task on Shakespeare dataset, the batch size $B = 32$ and learning rate $\eta = 0.2$ were similarly found and made constant for all methods under comparison.

We noticed that the reported accuracy could be further improved if deeper models are used for baselines as they can exhibit greater redundancy and thus benefit the performance of FedDrop. However, we are *not* pursuing state-of-the-art accuracy on these datasets but instead comparing FedDrop with other methods under fair experimental settings and validating its effectiveness. We also noticed that there exist larger alternative datasets (Hsu et al., 2020) for evaluation. Nevertheless, evaluating and reproducing all competing FL methods on such a large dataset is infeasible given the limited (university) computational budgets. Alternatively, we aim to conduct a fair empirical evaluation that demonstrate the effectiveness of FedDrop under a broad variety of scenarios.

Table 6: Layout of the model used for Fashion-MNIST training.

|  | Layer | Kernel | Stride | Feature shape | #Params | #FLOPs |
|---|---|---|---|---|---|---|
| 1 | Conv+ReLU | $5 \times 5$ | 1 | $32 \times 28 \times 28$ | 832 | 652 k |
| 2 | Max Pool | $2 \times 2$ | 2 | $32 \times 14 \times 14$ | — | 25.1 k |
| 3 | Conv+ReLU | $5 \times 5$ | 1 | $64 \times 14 \times 14$ | 51.3 k | 10.0 M |
| 4 | Max Pool | $2 \times 2$ | 2 | $64 \times 7 \times 7$ | — | 125 k |
| 5 | Conv+ReLU | $3 \times 3$ | 1 | $64 \times 5 \times 5$ | 36.9 k | 923 k |
| 6 | Avg Pool | $2 \times 2$ | 1 | $64 \times 2 \times 2$ | — | 1.6 k |
| 7 | FC | — | — | 512 | 132 k | 132 k |
| 8 | FC | — | — | 10 | 5.13 k | 5.13 k |
|  |  |  |  | **Total** | 226 k | 11.8 M |

Table 7: Layout of the model used for SVHN training.

|  | Layer | Kernel | Stride | Feature shape | #Params | #FLOPs |
|---|---|---|---|---|---|---|
| 1 | Conv+ReLU | $5 \times 5$ | 1 | $32 \times 28 \times 28$ | 2.43 k | 1.91 M |
| 2 | Max Pool | $2 \times 2$ | 2 | $32 \times 14 \times 14$ | — | 25.1 k |
| 3 | Conv+ReLU | $5 \times 5$ | 1 | $64 \times 10 \times 10$ | 51.3 k | 5.13 M |
| 4 | Max Pool | $2 \times 2$ | 2 | $64 \times 5 \times 5$ | — | 6.4 k |
| 5 | Conv+ReLU | $3 \times 3$ | 1 | $64 \times 3 \times 3$ | 36.9 k | 333 k |
| 6 | Avg Pool | $2 \times 2$ | 1 | $64 \times 2 \times 2$ | — | 576 |
| 7 | FC | — | — | 512 | 132 k | 132 k |
| 8 | FC | — | — | 10 | 5.13 k | 5.13 k |
|  |  |  |  | **Total** | 227 k | 7.54 M |

Table 8: Layout of the VGG-9 model used for CIFAR-10 training.

| | Layer | Kernel | Stride | Feature shape | #Params | #FLOPs |
|---|---|---|---|---|---|---|
| 1 | Conv+ReLU | $3 \times 3$ | 1 | $32 \times 32 \times 32$ | 896 | 918 k |
| 2 | Conv+ReLU | $3 \times 3$ | 1 | $64 \times 32 \times 32$ | 18.5 k | 18.9 M |
| 3 | Max Pool | $2 \times 2$ | 2 | $64 \times 16 \times 16$ | — | 65.5 k |
| 4 | Conv+ReLU | $3 \times 3$ | 1 | $128 \times 16 \times 16$ | 73.9 k | 18.9 M |
| 5 | Conv+ReLU | $3 \times 3$ | 1 | $128 \times 16 \times 16$ | 148 k | 37.8 M |
| 6 | Max Pool | $2 \times 2$ | 2 | $128 \times 8 \times 8$ | — | 32.8 k |
| 7 | Conv+ReLU | $3 \times 3$ | 1 | $256 \times 8 \times 8$ | 295 k | 18.9 M |
| 8 | Conv+ReLU | $3 \times 3$ | 1 | $256 \times 8 \times 8$ | 590 k | 37.8 M |
| 9 | Avg Pool | $8 \times 8$ | — | $256 \times 1 \times 1$ | — | 16.4 k |
| 10 | FC | — | — | 512 | 132 k | 132 k |
| 11 | FC | — | — | 512 | 263 k | 263 k |
| 12 | FC | — | — | 10 | 5.13 k | 5.13 k |
| | | | | **Total** | 1.53 M | 134 M |

Table 9: Layout of the model used for Shakespeare dataset training.

| | Layer | Shape | #Params | #FLOPs |
|---|---|---|---|---|
| 1 | Encoder | $80 \times 8$ | 640 | — |
| 2 | LSTM | $8 \times 256$ | 272 k | 275 k |
| 3 | Decoder | $256 \times 80$ | 21 k | 21 k |
| | | **Total** | 293 k | 296 k |

# F  ADDITIONAL EXPERIMENTAL DETAILS AND RESULTS

## F.1  TRADE-OFF CURVES

Figures 10 to 12 find the optimal configurations for each competing FL methods that produces their corresponding communication/computation trade-off curves. It is notable that FedDrop can extend the trade-off relationships much further and shows Pareto dominance against other algorithms under most conditions. In these figures, the configurations explored include combinations of FLOPs budget ratios $r \in \{0.25, 0.333, 0.5, 0.75\}$ and local epochs per round $E \in \{1, 2, 4, 8, 16\}$.

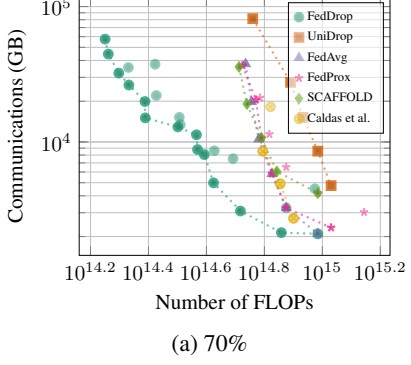

(a) 70%

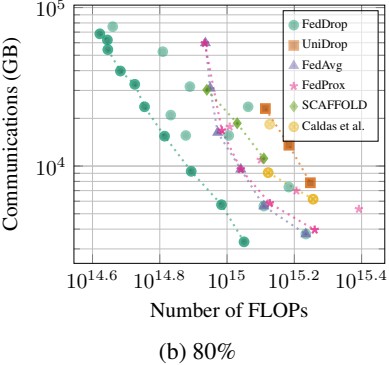

(b) 80%

Figure 10: The FLOPs *vs.* communication trade-off curves of different FL methods reaching a target accuracy for CIFAR-10. We highlight the Pareto optimal points with dotted lines.

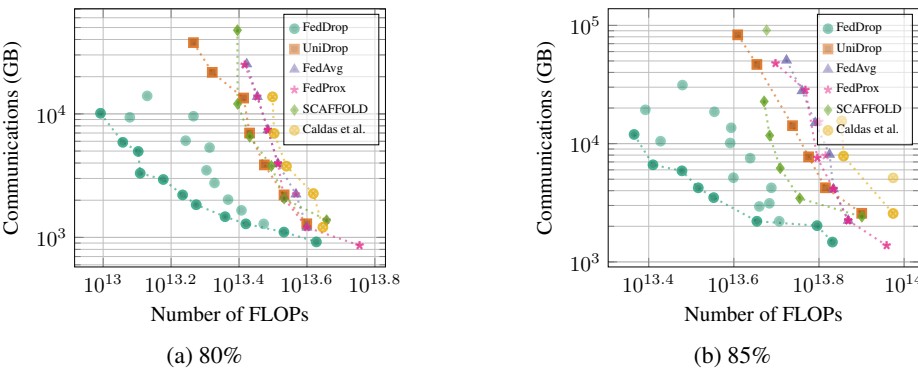

Figure 11: The FLOPs *vs.* communication trade-off curves for Fashion-MNIST.

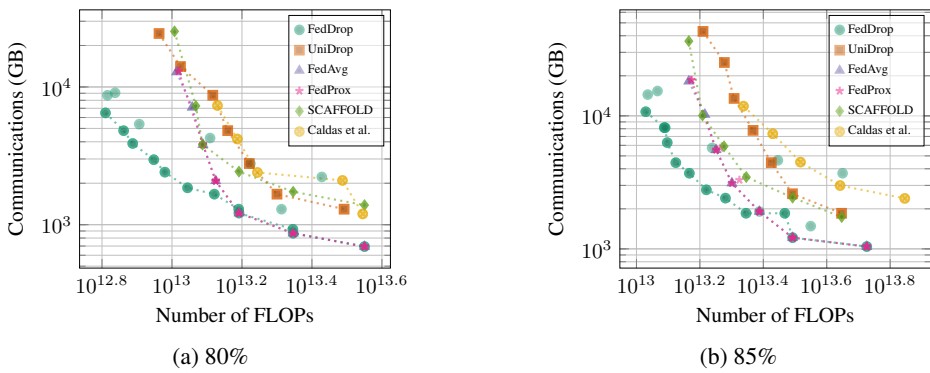

Figure 12: The FLOPs *vs.* communication trade-off curves for SVHN.

## F.2   DATA IMBALANCE

Figures 13 to 15 examine the effect of data imbalance on the performance of FedDrop. Again, it shows that FedDrop can manage the fastest convergence when compared to other methods, and an increase in data imbalance further enhances its effectiveness against the others.

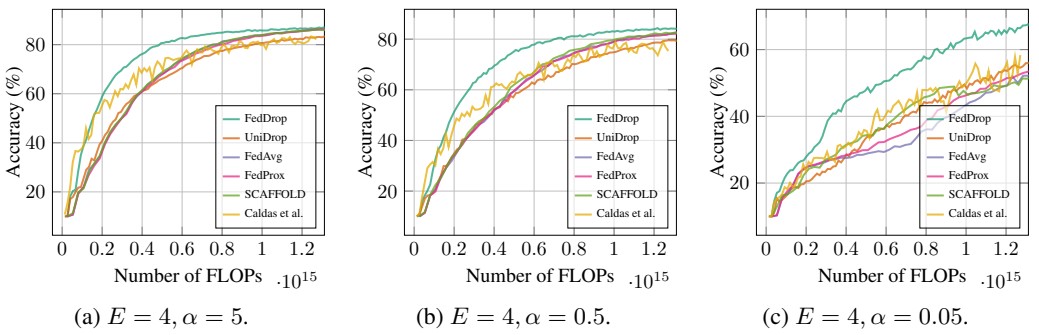

Figure 13: Varying $\alpha$ for the dataset splits $\mathcal{D}_{|\mathbb{C}|}(\alpha)$ of CIFAR-10.

## F.3   VARYING LOCAL EPOCHS PER ROUND

Figures 16 and 17 adjust the number of local epochs per round, and observe that with more frequent rounds (lower $E$), the accuracy gaps between FedDrop and the other methods become greater.

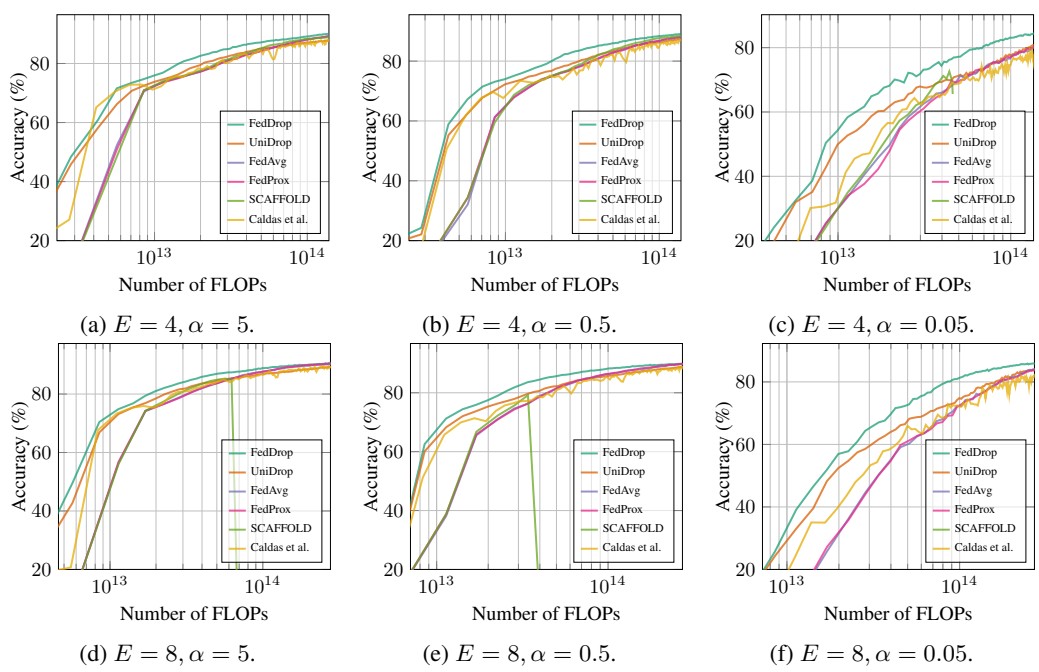

Figure 14: Varying $\alpha$ for the dataset splits $\mathcal{D}_{|\mathbb{C}|}(\alpha)$ of Fashion-MNIST. Log-scale is used for clarity.

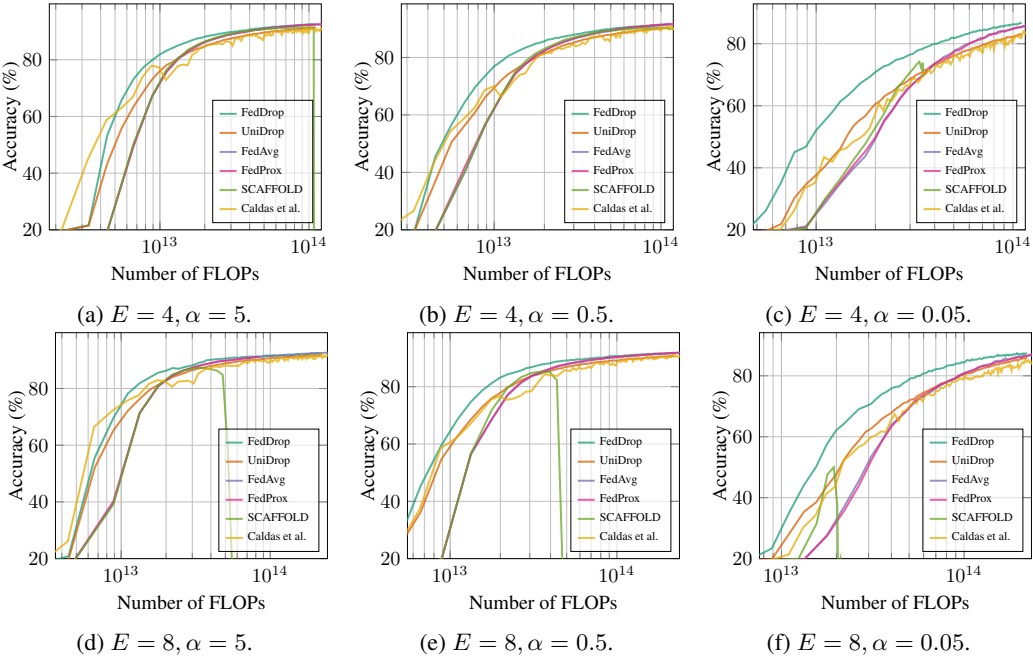

Figure 15: Varying $\alpha$ for the dataset splits $\mathcal{D}_{|\mathbb{C}|}(\alpha)$ of SVHN. Log-scale is used for clarity.

## F.4 VARYING FLOPS BUDGET RATIO

Figures 18 to 20 explore varying FLOPs budget ratios $r \in \{0.75, 0.5, 0.333, 0.25\}$. We found that a higher sparsity (lower ratio) can improve the overall convergence rate *w.r.t.* FLOPs expended. However, the final converged accuracy may degrade; this can be fixed by simply adjusting the budget ratio to approach 1.

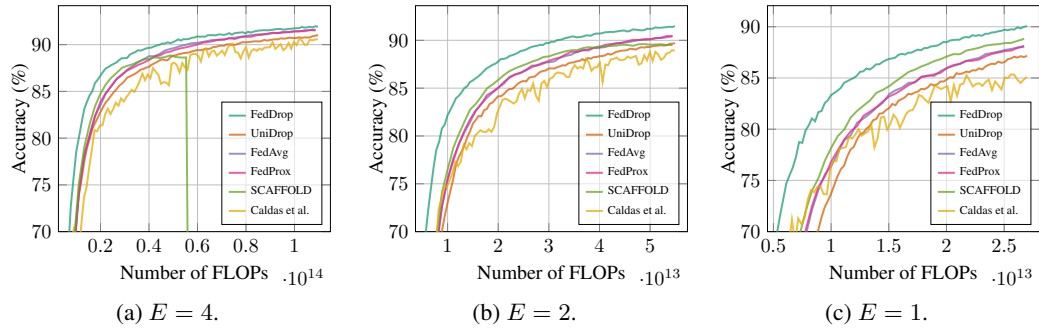

Figure 16: Comparing FL methods with the same local training epochs per round $E$ used for SVHN.

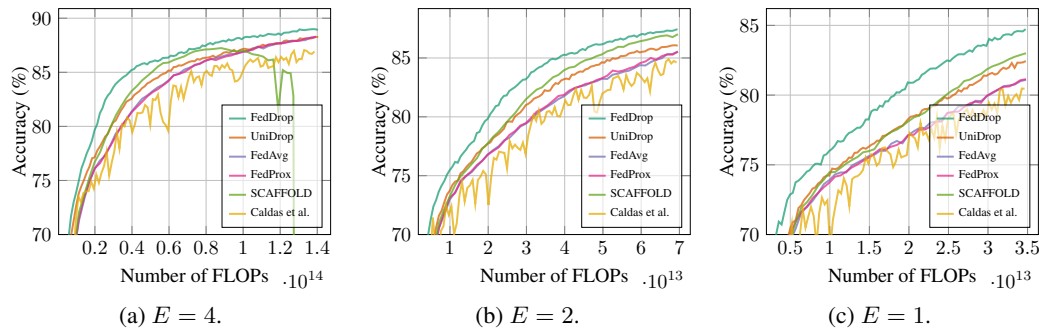

Figure 17: Comparing FL methods with the same local training epochs per round $E$ used for Fashion-MNIST.

### F.5 FRACTIONAL DEVICE PARTICIPATION

We increase the number of all devices to 1000, and reduced the fraction of participating devices per round to $1\%$ for each of the datasets. All competing methods perform uniform device subsampling each round as this was commonly reported for their original experiments McMahan et al. (2017); Li et al. (2020); Karimireddy et al. (2020); Caldas et al. (2018a). For FedDrop to perform well, we uniformly sample devices every 2 rounds, to train with freshly updated probabilities. In Figure 21, Figure 22, Figure 23 and Figure 24, we compare FedDrop against all competing methods, and show that it can bring notable improvements in both communication and computation efficiencies.

### F.6 ACTUAL TRAINING RUN TIME

In Table 10, we provide the average training times per image on an iPhone 12 Pro running the Fashion-MNIST model with a batch size of 4. Here, each result was obtained under optimal conditions where no frequency throttling occurred. The structure of the model can be found in Table 6 of Appendix E. Note that we were only able to report CPU run times, as Core ML (cor, 2021) currently only support training on CPUs. We implemented a custom convolutional layer to take advantage of the sparse input and output feature maps.

Table 10: iPhone 12 Pro run time of training on CPUs.

| FLOPs ratio | Theoretical speed-up | Average run time per image (μs) 5 runs | | | | | Average (μs) | Actual speed-up |
|---|---|---|---|---|---|---|---|---|
| 1.0 | 1× | 966.64 | 867.34 | 832.46 | 1072.06 | 1178.76 | 983.45 ± 128.50 | 1.00× |
| 0.5 | 2× | 544.40 | 551.54 | 567.80 | 568.44 | 576.32 | 561.70 ± 11.83 | 1.75× |
| 0.25 | 4× | 388.04 | 372.54 | 378.06 | 380.22 | 379.40 | 379.65 ± 4.98 | 2.59× |

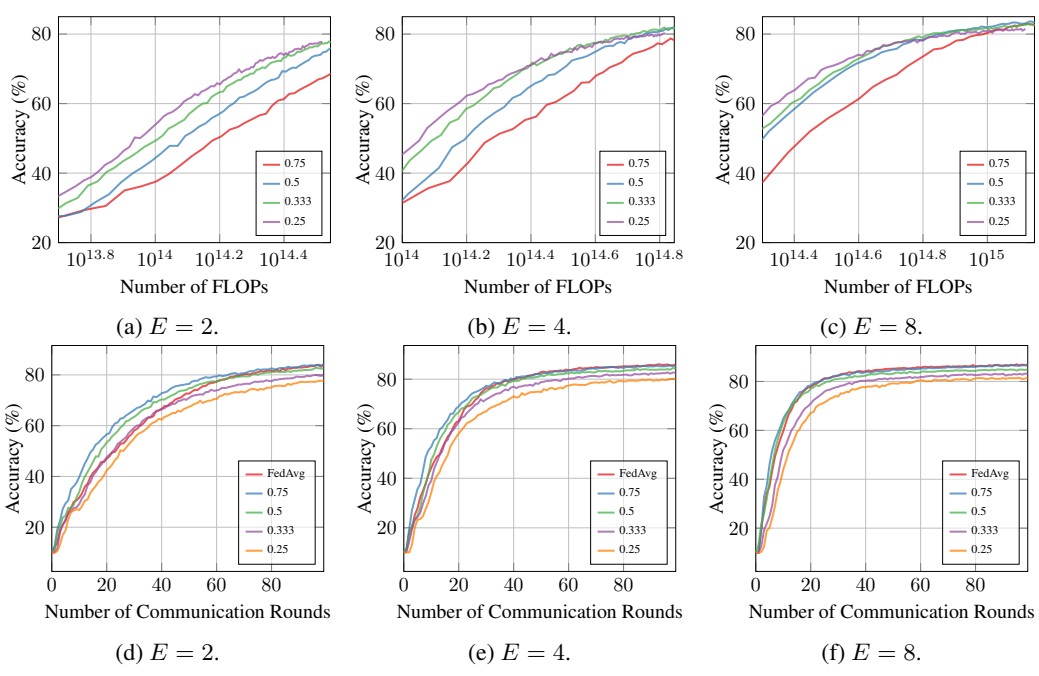

Figure 18: Reducing the FLOPs budget ratio $r$ for CIFAR-10.

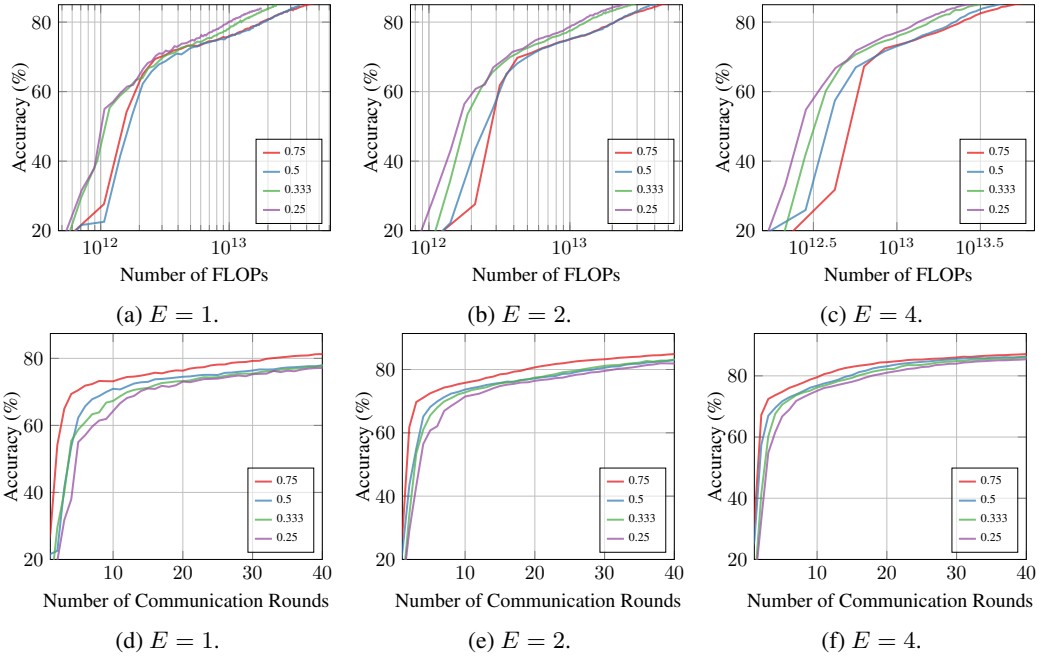

Figure 19: Reducing the FLOPs budget ratio $r$ for Fashion-MNIST.

For the FedAvg motivation example in Section 1, we assumed 20 clients, with 2500 examples per epoch on each client, and 4 training epochs per round, which is in line with our experiments with 20 devices. We extrapolate the results to 100 epochs for a batch size of 4 to estimate the run time to be $983.45\,\mu\mathrm{s} \times 2500 \times 4 \times 100 = 16.39\,\mathrm{min}$ without frequency throttling. As each parameter is stored as a 32-bit float, each client would upload and download a model of size $226\,k \times 4\,\mathrm{B} = 904\,k\mathrm{B}$ once per round; the overall transmission per client is thus evaluated to be $904\,k\mathrm{B} \times 2 \times 100 = 176.56\,\mathrm{MB}$.

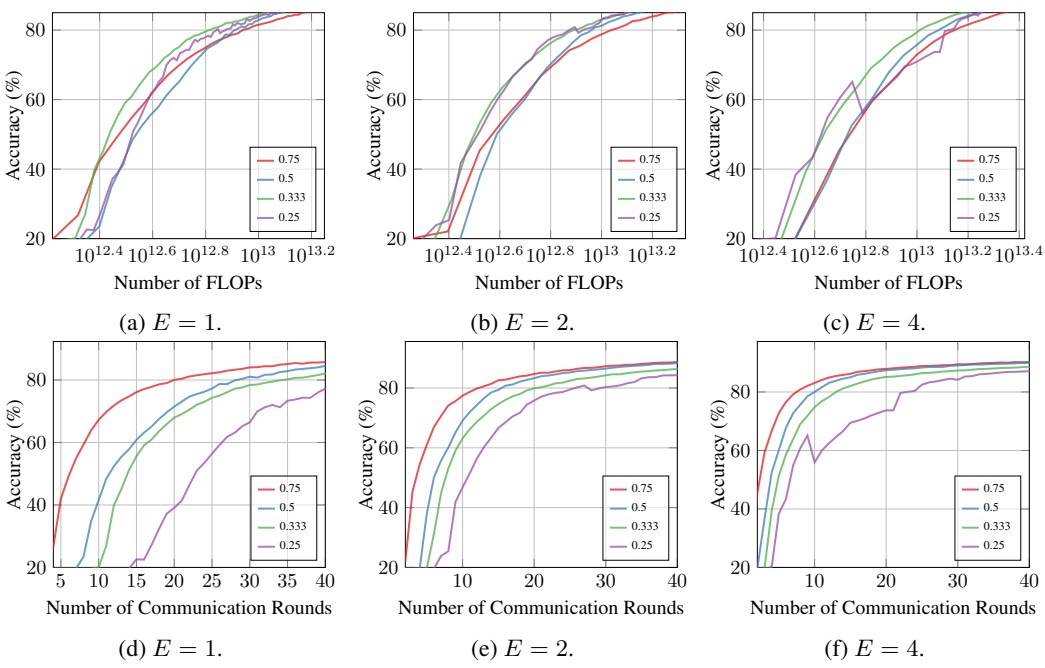

Figure 20: Reducing the FLOPs budget ratio $r$ for SVHN.

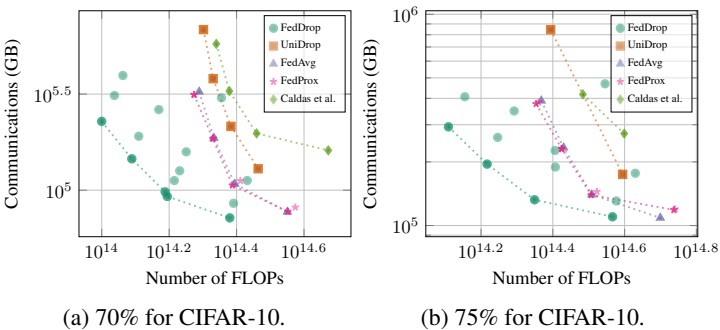

Figure 21: Comparing the FLOPs *vs.* communication trade-off across different FL methods reaching a target accuracy for CIFAR-10. We highlight the Pareto optimal points with dotted lines.

## F.7 ADDITIONAL RESULTS ON THE MOTIVATING EXAMPLE

For reference, in Figure 25 we included all parameter update magnitudes of the first convolutional layer for the motivating example given in Figure 1 in Section 1.

## F.8 RESULTS ON THE NLP TASK

We provide results on Shakespeare dataset for full client participation in Figure 26 to Figure 28 and fractional client participation in Figure 29 to Figure 31. The experiment configurations are introduced in detail in Appendix E.

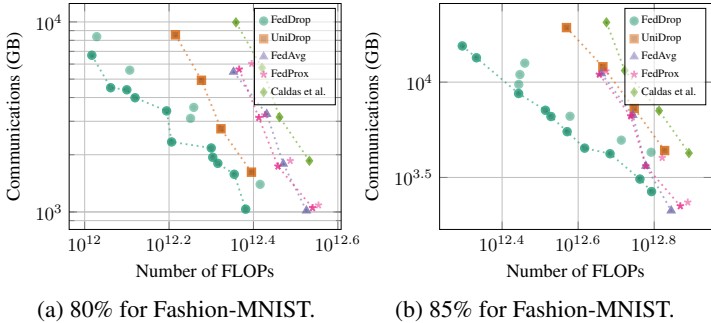

(a) 80% for Fashion-MNIST.   (b) 85% for Fashion-MNIST.

Figure 22: Comparing the FLOPs *vs.* communication trade-off across different FL methods reaching a target accuracy for Fashion-MNIST. We highlight the Pareto optimal points with dotted lines.

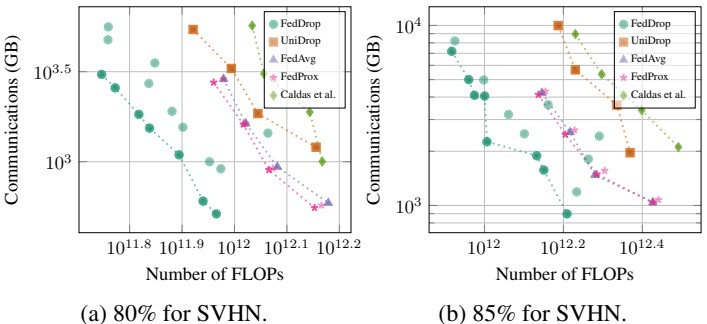

(a) 80% for SVHN.   (b) 85% for SVHN.

Figure 23: Comparing the FLOPs *vs.* communication trade-off across different FL methods reaching a target accuracy for SVHN. We highlight the Pareto optimal points with dotted lines.

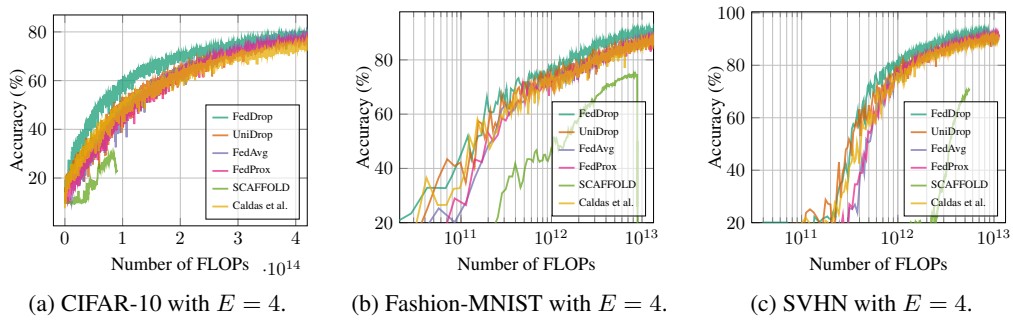

(a) CIFAR-10 with $E = 4$.  (b) Fashion-MNIST with $E = 4$.  (c) SVHN with $E = 4$.

Figure 24: Reducing device participation to 1%. Some plots use log-scale for clarity.

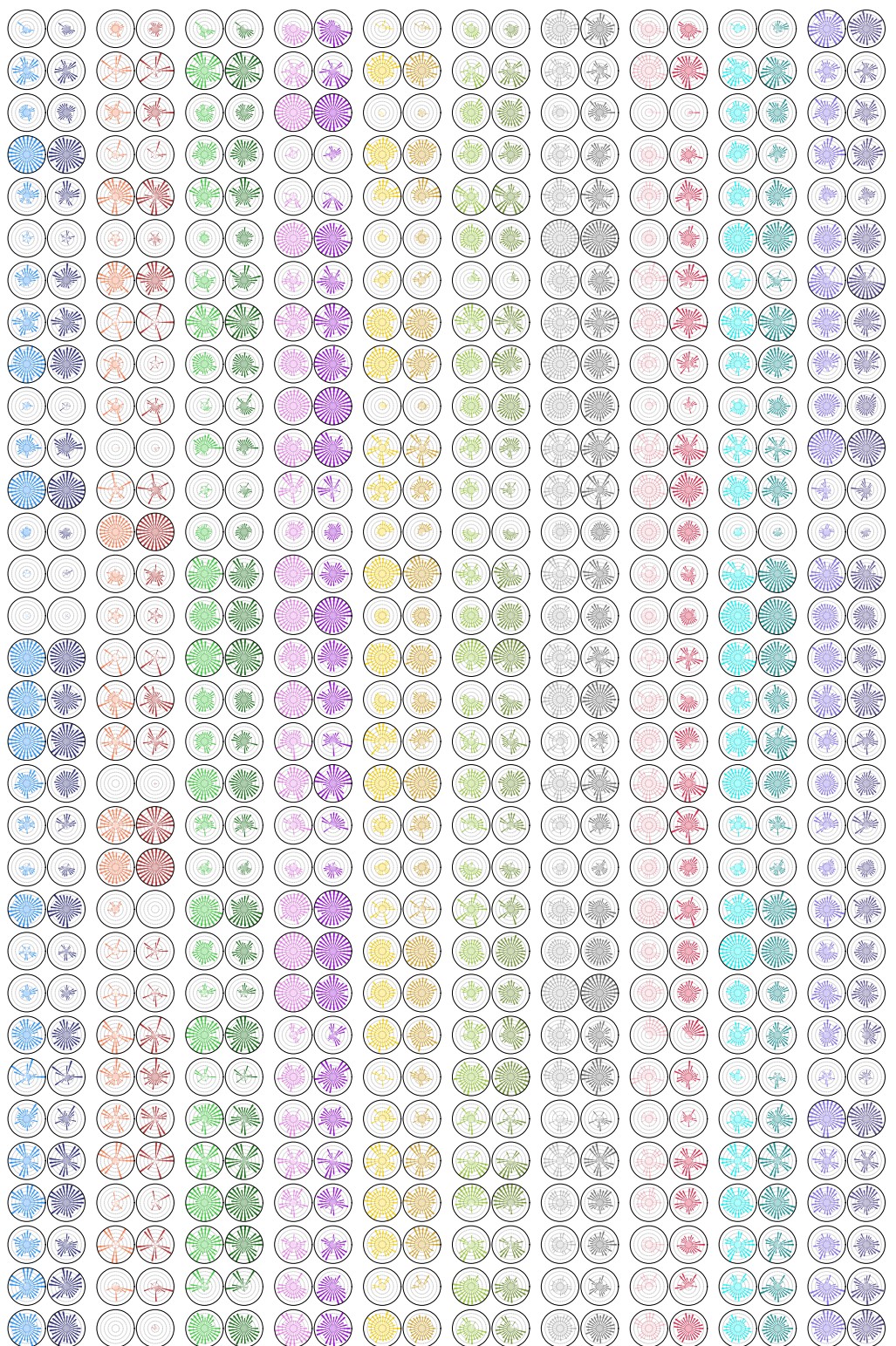

Figure 25: All parameter update magnitudes for each channel (row) in the first convolutional layer for all clients (column) after 1 round of training. The pairs of clients with same colors (one lighter and one darker) signify that they shared the same image class. The length of each ray is the magnitude of the parameter update, and the parameters are ordered by the angles of the rays. The results are taken from the motivating example in Figure 1.

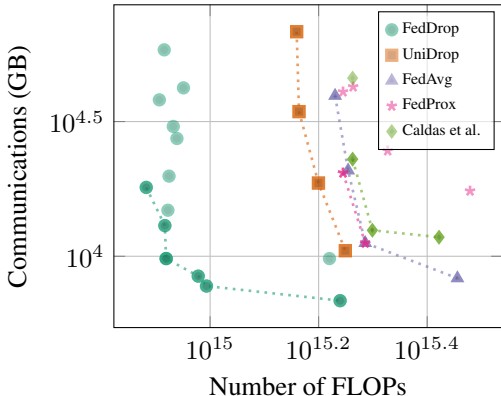

Figure 26: Comparing the FLOPs *vs.* communication trade-off across different FL methods reaching a 50% accuracy for the Shakespeare dataset. We highlight the Pareto optimal points with dotted lines.

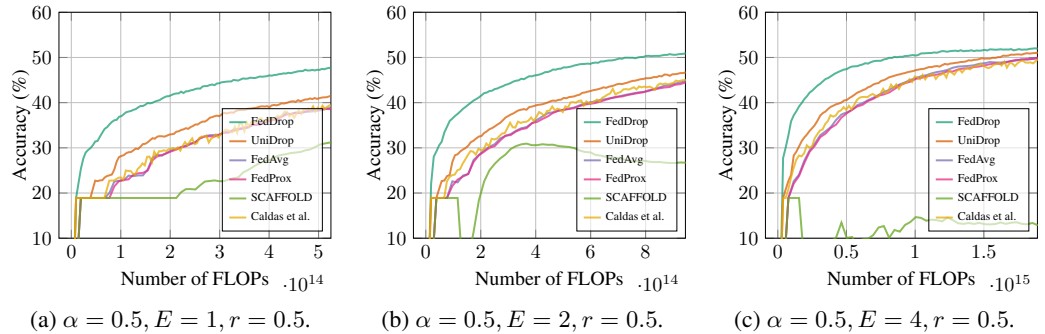

(a) $\alpha = 0.5, E = 1, r = 0.5$.

(b) $\alpha = 0.5, E = 2, r = 0.5$.

(c) $\alpha = 0.5, E = 4, r = 0.5$.

Figure 27: Comparing FL methods with the same local training epochs per round $E$ used for Shakespeare.

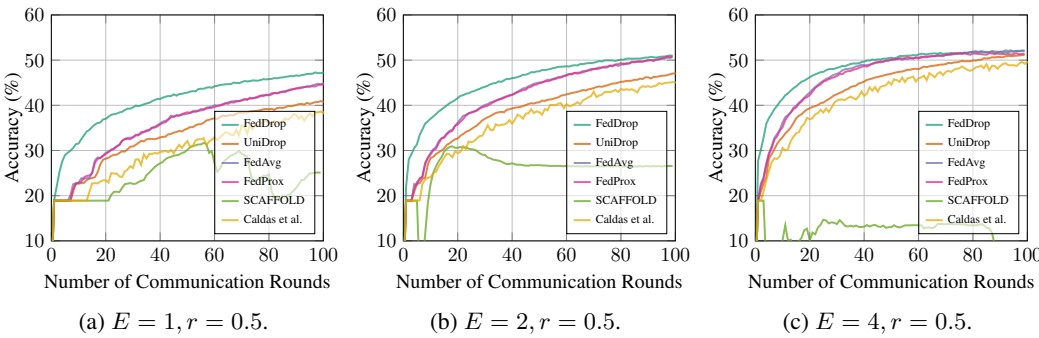

(a) $E = 1, r = 0.5$.

(b) $E = 2, r = 0.5$.

(c) $E = 4, r = 0.5$.

Figure 28: Comparing FL methods on the number of communication rounds *vs.* accuracy with the same local training epochs on Shakespeare.

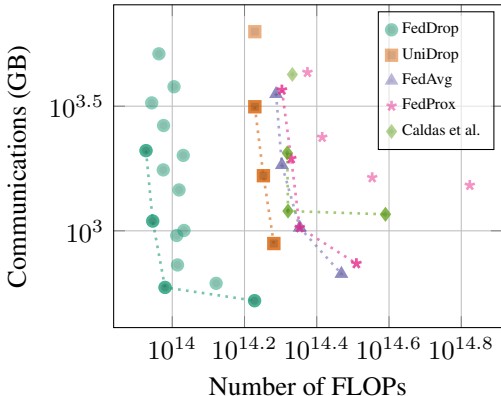

Figure 29: Comparing the FLOPs *vs.* communication trade-off across different FL methods reaching a target accuracy for a given dataset. We highlight the Pareto optimal points with dotted lines.

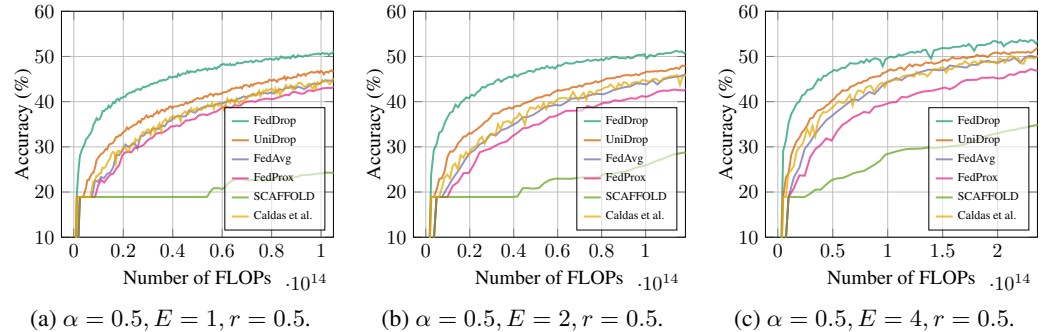

(a) $\alpha = 0.5, E = 1, r = 0.5$.    (b) $\alpha = 0.5, E = 2, r = 0.5$.    (c) $\alpha = 0.5, E = 4, r = 0.5$.

Figure 30: Comparing FL methods with the same local training epochs per round $E$ used for Shakespeare.

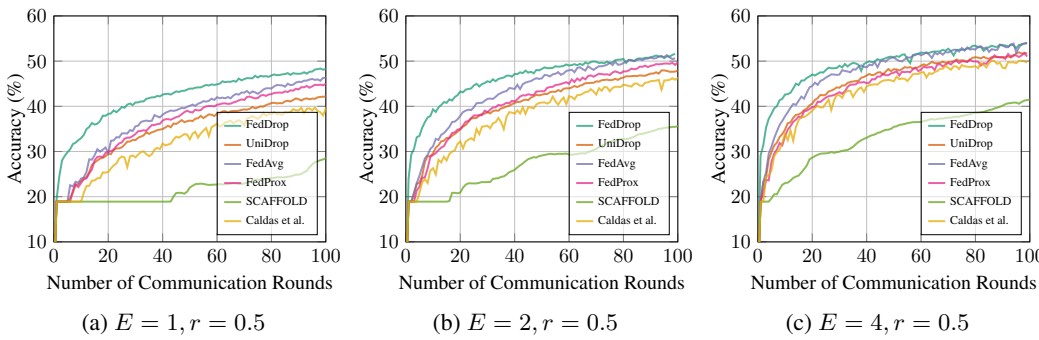

(a) $E = 1, r = 0.5$    (b) $E = 2, r = 0.5$    (c) $E = 4, r = 0.5$

Figure 31: Comparing FL methods on the number of communication rounds *vs.* accuracy with the same local training epochs on Shakespeare.

