# OpenReview forum: "FedDrop: Trajectory-weighted Dropout for Efficient Federated Learning"
_ICLR.cc/2022/Conference — ICLR 2022 Submitted_

### Official Review · Reviewer_A6bP · 2021-10-25

**Correctness:** 4
**Technical Novelty And Significance:** 3
**Empirical Novelty And Significance:** 2
**Recommendation:** 5
**Confidence:** 3

**Main Review:**

Strengths
- The paper tackles the problem of computational cost to the clients, which is relatively understudied compared to the communication costs.
- The technique does not make any simplifying assumptions and is applicable to real world FL use cases.

Weaknesses
- The method proposed comes with increased communication costs to the client. While the authors show that they can favourably trade communication costs for computational costs, generally practitioners are more concerned with communication costs, as these can impose real costs on the users (such as using up limited wifi resources), whereas computational costs are largely in training time which are 'paid' by the server.
- The results here are very specific to computer vision. Would similar results be possible for other problems such as NLP?
- The log graphs in figure 5 are odd and challenging to read, with different spacing applied to the x and y axis.
- In figure 5.c how is FedDrop able to achieve lower communication costs than FedAvg?
- The results in table 1 are a little difficult to interpret. If I understand them correctly, they indicate the computational cost of different method to reach a given accuracy, under the constraint that they keep below a certain communication budget. If this is correct, it would be much more valuable to the reader to record the actual communication costs of each method as well as their computational costs. Simply saying that all methods stayed below some budget may be hiding important information about the additional communication costs of FedDrop.
- Empirical results appear quite weak, with a 10x - 100x increase in the communication costs resulting in less than a 10x reduction in computation costs.

**Summary Of The Paper:**

The authors propose a new method of coordinated, per client and weight dropout for federated learning. The intuition behind the method is to increase the dropout probability (probability of having a weight set to zero) for weights where different clients often have opposite parity gradients (as when those gradients are summed together are likely to cancel each other out). The authors show that doing this can reduce the computational cost on clients of FL by up to 3x.

**Summary Of The Review:**

While reducing client computational costs is an important task in FL, the method proposed in the paper comes at a very high communication costs, which makes it unlikely to be practically useful while communication costs are still the primary concern in FL.

---

> ### Author Response · Authors · 2021-11-15
> **Thanks for the in-depth review, concerns addressed below.**
>
> We would like to address your itemized concerns:
> 1. We would like to point out that it is difficult to quantify computation and communication costs in a unified metric, as it is highly dependent on the environment. For this reason, we rely on proposing a trade-off relation between the two resources and designed FedDrop to dominate in terms of both. It is also surprising to us that it was mentioned in the review “computational costs are largely in training time which are ‘paid’ by the server”. Could you provide us with examples that incentivize the users to contribute compute for FL?
> 2. We are running additional results using RNN models with NLP tasks and we hope to update them for you soon.
> 3. Because of computational constraints we have for running experiments, we were only able to explore the trade-off curves in limited ranges of hyper-parameter choices. This explains why trade-off axes do not have the same proportion.
> 4. Please take a look at Figure 18, and it shows that FedDrop can actually out-perform FedAvg in terms of the number of rounds in some experiments. The communication overhead of FedDrop is almost negligible, as it additionally transmits a few hundred probability values per round. This explains why FedDrop can sometimes transmit a smaller amount of bytes than other methods to achieve the same accuracy with the same set of hyper-parameters.
> 5. Each algorithm was run multiple times with different hyper-parameters, and we report the minimum number of FLOPs required to achieve the target accuracy under the same communication budgets. We are aware that this may not provide a complete picture (as there is actually a 3D trade-off between accuracy, computation and communication which is difficult to visualize on paper), so we provide 2D slices of the picture with tables and trade-off curves.
> 6. While it is true that the slope between communication and computational costs may not be close to 1 in the trade-off figures, this can be attributed to the fact that the ranges of hyper-parameters explored are limited by our computational resources (as mentioned in 3 above). The emphasis of FedDrop is not on the specific choice of hyper-parameters that provide a 1:1 slope between the two costs, but is to provide a better trade-off when compared to other methods, regardless of the choice of communication/computation budget ratios.

---

> > ### Comment · Reviewer_A6bP · 2021-12-01
> > **Comments**
> >
> > 1) Sorry I should have been more clear here: By “computational costs are largely in training time which are ‘paid’ by the server” I meant the following: In FL client compute is generally only used when it is basically free to the client (that is when the device is idle and connected to external power). Therefore the greatest 'cost' from additional client compute is in the time it takes the client to run the computation and return the result to the server. But this cost impacts the server (in that it takes more overall time to train), so reducing it only benefits the server. In contrast, the communication costs to the client can be non-trivial, as even a few tens of MB is a non-zero fraction of common bandwidth allowances before additional charges/throttling etc is applied.
> >
> > 2) Thank you these results agree with the previous ones which is good.
> >
> > 3) I would suggest plotting these with more standard axes as these are difficult to interpret.
> >
> > 4) I see thank you.
> >
> > 5) In this case, it would be useful to actually provide how much of the budget was used in that experiment, along with the budget. Just providing the threshold it's under makes it difficult to compare methods.
> >
> > 6) Providing a better trade-off than other methods is important, but the actual values of the trade-off are also important, and with the current values of the trade-off it's unclear when anyone would want to be anywhere but the communication-efficient end of the parato curve.

---

> > > ### Author Response · Authors · 2021-12-01
> > > **Thank you for the comments.**
> > >
> > > We would like to thank the reviewer for the comments and address your remaining concerns.
> > >
> > > 1: Thanks for the clarification, and I appreciate and understand your reasoning. If it is reasonable to assume that power is not the constraint if the device only trains when connected to the external power, then it is also fair to assume that Wi-Fi with unlimited data allowance is also pervasive. Hypothetically, the user may be pretty annoyed if they find out that their device was not charing because it is training on their personal data, or may be confused to find their phone as hot as an active oven. Even if we were to think that energy and communication cost are both negligible, time is still an important factor to minimize and is dependent on both the computational and communication costs in a non-trivial platform-dependent way. Finally, FedDrop dominates other methods in terms of communication vs. computation, i.e. it can often reduce FLOPs under the same communication costs.
> > >
> > > 3: Please see 6 below.
> > >
> > > 5: Thank you for the suggestion, we can certainly update the table to reflect the budgets.
> > >
> > > 6: We appreciate for bringing to our attention that the experiments placed greater weight on computational costs. Unfortunately, we had limited compute budget and had no way to know this before producing the Pareto curves. We would like to point out that the union of the sets of experiment setups from previous work (FedAvg, FedProx and SCAFFOLD) provided the basis for the range of hyper-parameters explored in the Pareto curves.

---

### Official Review · Reviewer_wCnx · 2021-11-02

**Correctness:** 2
**Technical Novelty And Significance:** 2
**Empirical Novelty And Significance:** 2
**Recommendation:** 3
**Confidence:** 4

**Main Review:**

- The premise of the paper that other works actually over-emphasize communication, while local computation is as time-consuming is a bit misleading. I agree with the fact that the act of communicating information, i.e. sending the updated model parameters over the channel might not take so long. But this is not the only factor defining the time that each communication round takes in FL. There is a lot of overhead associated with each FL round: 1) select an available device (usually connected to wifi and power) 2) make sure the device does not become unavailable,... 3) collect enough updates from enough devices each with their own delays (depending on the loads, computation power...); note that here we are constrained by the slowest device. All these factors actually add a lot of overhead to each FL round. So just looking at the communication piece without seeing this bigger picture is misleading. In fact, all the methods that are compared against are targeting the reduction of the number of rounds and not communicated bits; if communicated bits were the target, the authors should have compared against methods that directly target that, e.g. by compressing the updates.
- The theoretical results that are presented by the paper are very weak. Their statements are misleading/sloppy or wrong and lack assumptions. And to be honest the theorems should not be called that.
  - Theorem 1: This requires Assumptions 1-2 which are very strong and definitely not true for all the layers. And the result under these assumptions is trivial.
  - Theorem 3 is just obvious!
  - In Lemma 1, there is no reasonable justification to drop the Hessian term. It cannot be justified and all the approximations based on that are not valid. Note that Lemma 2 and Theorem 2 are built on top of the closeness of this approximation! Finally, although the results are presented for gradients, there is a big leap when using this in practice, where the updates are actually the accumulation of multiple steps instead of gradients.
- The algorithm is impractical and possibly not very useful in many cases:
  - No secure aggregation can be integrated into it, while many FL systems require secure aggregation.
  - The steps on adjusting the drop-out probabilities are computation/memory/communication intensive. First of all the computation there scales with the number_parameters x batch_size^2. which with large batch sizes in practice could be too much. More importantly, it will require the server to track p's for all the clients which is not possible in large-scale applications. The other alternative is that the clients keep sending the server their p's which will increase the communication as the size of p's is large. In fact, I suspect for large-scale realistic problems with many clients and a low percentage of participation per round (which are mostly absent from the experiments) this stateful situation would not necessarily lead to any benefit (as the clients' p's are updated so infrequently). I believe that is why in the case where there are 1000 clients and 1% participation the authors have had to resort to choosing the same clients for two consecutive updates; see appendix F.5. One possible issue with this is that it is not possible to make sure that the same set of clients are available for two consecutive updates due to external situations (clients may become temporarily unavailable at any point during the training).
- The experimental results are not convincing and I do not think the gains are significant enough.
  - The method is only tailored and only tested on CNNs while many FL problems do not use them. This has made the experimental section less convincing as there are many usual FL benchmarks that are absent.
  - Flops are not easy to estimate and most of the reported results are in terms of FLOPS. While even the authors' estimates of actual FLOPS savings show that the theoretical numbers used for FLOPS (as the x-axis in many figures) could be as much as 1.5x worse; see table 9.
  - There are only a few plots comparing the rounds, while that is important. In fact, there is actually only one plot (Fig 18) that compares FedDrop with another method FedAvg over the number of rounds. And in that plot, it is clear that FedDrop cannot achieve the final accuracy of FedAvg with low r (which is the setting that has gains based on other plots). And in order to achieve similar final performances to FedAvg, FedDrop has to run with r=0.75 or higher which does not seem to result in any significant gain (in other metrics).
  - I noticed that the results that show a significant gain for FedDrop, e.g. Fig 4, actually use very low E; compare Fig 4 to Fig 8, where E=8. Note that large E is essential in getting good results in FL. Also, this is very unfair to other methods as their only tool in reducing the communication rounds is local computation; while FedDrop is additionally using r! Based on this I do not believe the authors have done a fair comparison. It is worth noting that in many of the plots it seems that running the other algorithms for a longer time could also result in better results than FedDrop (e.g. see Fig 15); Interestingly, even in this case Fig 16 shows a large gap in favor of FedDrop which is due to the use of low E as well! Unfortunately, I do not think the authors use good criteria to stop the algorithms (e.g. when they have converged).

======= After Rebuttal =======

I do not think the authors' response addressed my concerns.
- Around the theoretical results: my original comment was around the fact that just stating some theoretical results with assumptions that are not realistic or verified does not add to the paper's value. Moreover, the results in Theorem 3 are obvious. And the term Theorem is reserved for significant contributions and does not apply here. Also, the authors did not provide any meaningful justification for the approximation in Lemma 1 (and the divergence from theory where there are multiple local steps).
- For the theoretical results: As other reviewers have also mentioned, the empirical results are quite weak. And as I mentioned in my original review they are not fair (not running the algorithms for long enough, not allowing larger E for other methods, ...). Unfortunately, the authors did not properly address these issues in their rebuttal. Please refer to my original review for detailed issues regarding the empirical results in the paper.

**Summary Of The Paper:**

The paper proposes FedDrop and a synchronized drop-out strategy to reduce local computation during the training of FL models.

**Summary Of The Review:**

Based on the above comments, I do not believe this paper passes the acceptance threshold. I encourage the authors to address the points around: motivation, practicality, and rigorous experimental comparisons with other works.

---

> ### Author Response · Authors · 2021-11-15
> **Thanks for the detailed review, we would like to address your concerns below. (1/2)**
>
> 1. We would like to address the concerns on the evaluation of communication costs below:
> 	1. We would like to point out that FedDrop does not compress communication, and replacing the communication costs with rounds would still place FedDrop in the lead even in terms of rounds-FLOPs trade-offs in all of our experiments. We only chose communication bytes for comparison as some of the competing methods may noticeably compress (e.g. Caldas et al.) or incur additional overhead (e.g. SCAFFOLD has a 2x overhead) in communication costs.
> 	2. We agree that there are additional overheads associated with communication besides the amount of transmitted data, they are however difficult to quantify and would be highly dependent on the specific details of the environment. We believe the accumulated communication bytes is most suited for the evaluation of communication costs while being agnostic to the platforms.
> 	3. Algorithms that compress communication bytes are not pertinent to the comparisons for the reason explained above in 1. We did not choose to incorporate compression in this work as it dilutes the main narrative (i.e. computational saving), while it is certainly possible to do so in a fairly straightforward way.
> 2. For the theoretical results:
> 	1. We would like to point out that Assumptions 1 and 2 are from the highly cited [1] that introduced He initialization, which is implemented by PyTorch in [torch.nn.init.kaiming_uniform_](https://pytorch.org/docs/stable/nn.init.html). While the assumptions may not be true in the general case, it is practical in constraining the initialization conditions for feed-forward NNs to derive the result in Theorem 1.
> 	2. We are happy to simplify the proofs of Theorems 1 and 3. We appreciate if the reviewer could provide suggestions to simplify them further, besides stating they are trivial and obvious.
> 	3. The Hessian term is often omitted from many of the commonly used machine learning algorithms (e.g. SGD) to trade convergence rate with computational costs. In practice, this approach is highly effective as demonstrated by our experiments.
> 3. For the practicality of the algorithm:
> 	1. As the server requires only the trajectory-similarity tensor, where its elements can be approximated by the dot-products of model updates projected onto a lower-dimension space with random bases, this can also prevent the server from inferring data from the client models.
> 	2. The time complexity of optimization of dropout probabilities does not involve the number of parameters and the batch size. In Appendix D.2, we provide an in-depth discussion of the computational overhead of the optimization objective. To summarize,
> 		* The optimization has a complexity of $O(INC^2)$ where $I$ is the number of optimization iterations, $N$ is the number of channels, and $C$ is the number of participating clients.
> 		* The size of the $p$ transmitted per round is actually very small, which is $N$ values for each client, giving a total of $N \times C$ values.
> 		* In our test case (Appendix D.2), clients consume 1.29 T FLOPs per round (50% of FedAvg under $r = 0.5$), and the server overhead is ~2% of the total computation.
> 		* The reason for choosing the same clients for consecutive updates is to actually simulate to some extent the behavior of client joining and leaving. May we suggest that it could also be unrealistic to expect each round to always start with an entire set of freshly joined random clients.
> 		* The clients would consume multiple hours of training in a realistic implementation on an iPhone 12 Pro (extrapolating from the profiling result in Table 9), whereas the server side only requires minutes to complete on an Nvidia V100 GPU. This provides a strong case for trading a fraction of server computation for much reduced client computation.
>
> [1]: Kaiming He, et al., Delving Deep into Rectifiers: Surpassing Human-Level Performance on ImageNet Classification, https://arxiv.org/abs/1502.01852

---

> ### Author Response · Authors · 2021-11-15
> **... Continued response to address your concerns below. (2/2)**
>
> 4. For concerns related to the experimental results:
> 	* We are running additional results using RNN models with NLP tasks and we hope to update them for you soon.
> 	* In fact, actual FLOPs and theoretical FLOPs are almost identical, please see Appendix D.5 and Table 5 for a discussion. In terms of run time, a gap between actual and theoretical speed-ups is expected, and could be highly dependent on the implementation and platform chosen. Because of library API limitations, our implementation incurs additional overhead of parameter movements, which could be circumvented by implementing directly in C or CUDA. However the work involved is substantial and irrelevant to the contribution of this paper. Finally, sparsified models also bring additional memory reduction (see Appendix D.4 and Table 4), which is often desirable for resource-constrained devices.
> 	* It is noteworthy that FedDrop is not a method to minimize the number of rounds directly, but to significantly reduce computation in terms of the number of FLOPs by expending more rounds. In Figures 10-12 and Figures 21-23, it can be observed in most cases FedDrop dominates FedAvg in the trade-off between computation (FLOPs) and communication (GB). Moreover, note that both FedDrop and FedAvg do not compress communicated bytes and this cost is proportional to the number of rounds used. Typically for the all competing FL algorithms, this means that asymptotically approaching the highest accuracy would consume exponentially more resources. To achieve highest accuracy regardless of budgets, it would be best to increase $r$ gradually to 1; when $r=1$, it is identical to FedAvg. The goal of FedDrop is not to improve the baseline accuracy given unlimited budget, but is to find the best model under resource constraints.
> 	* For all competing algorithms we explore their communication and computational trade-offs by sweeping $E$ and the batch size $B$ (as shown in Figures 10-12 and Figures 21-23), and find their respective Pareto optimal curves. It is notable that for Caldas et al. and UniDrop, we also allow $r$ to adjust their sparsity, and FedProx allows a varying $\mu$. Therefore, only FedAvg has no additional hyper-parameter. If you believe it is unfair to compare the methods with the additional optimization of $r$, we can certainly produce additional plots with $r$ fixed to a constant. Regarding the stopping criteria: as mentioned above, the stopping criteria for all of the experiments are always resource constraints, i.e. identical number of rounds, communications or FLOPs budgets.

---

> ### Author Response · Authors · 2021-12-01
> **Responses to the reviewer's after rebuttal update.**
>
> Did our response addressed your concerns regarding the premise of the paper and the practicality of the algorithm? We did not receive further questions regarding them.
> In this case, we believe your concerns are partially addressed with the following remaining questions that we would like to provide answers below.
>
> 1. Regarding the reviewer's original comment on the assumptions, please engage with us in our response in 2.1, and let us know if it addressed your concerns, if not then please let us know the reasons and we will answer your further questions timely. We believe the extensive amount of experiments (Figures 4-31 and Tables 2-3) should provide strong empirical evidence for the approximation used in Lemma 1 and multiple steps. In addition, if the reviewer objects the use of the term "Theorem" to state the contribution, then we are happy to change it. Again, we would like to hear from you on how it can be changed besides stating that the proof is obvious. Please provide suggestions to us and we would appreciate it greatly.
> 2. We would like to engage in a discussion in how our experiments are unfair to other competing methods. We can start by responding to your newly stated concerns:
>     * Running for the best accuracy would mean large variances in computational and communication budgets, as it is often observed that the test accuracy may spike to a new high by random chances, it would be difficult to attribute the resources consumed to either random chances or actual improvements by the method.
>     * Running for the best accuracy is also is unrealistic, as it would simply mean unlimited budget. Unlimited budget is not relevant to the motivation of the paper. In this case, FedDrop would be identical to FedAvg as there is no need for sparsity.
>     * The range of the numbers of local epochs $E$ chosen $\\{1, 2, 4, 8, 16\\}$ encloses the ranges of the originally reported experiment setups we compared against (FedAvg, FedProx, SCAFFOLD). Note that larger values of $E$ would only result in a greater increase in computational costs.

---

### Official Review · Reviewer_guLZ · 2021-11-02

**Correctness:** 4
**Technical Novelty And Significance:** 2
**Empirical Novelty And Significance:** 2
**Recommendation:** 5
**Confidence:** 4

**Main Review:**

The papper consists of following strengths and weaknesses

Strengths:
- The dropout mechanism is simple yet effective in deep learning. It is interesting to see how to use dropout to design new federated learning algorithm.
- Extensive numerical experiments verify the advantage of FedDrop.

Weaknesses:
- The paper lacks discussion on the convergence of the algorithm.
- The paper only shows experiments with neural network examples. I wonder how the methods perform with models that are not neural networks.
- The algorithm requires to solve an extra sub-optimization problem. The cost of solving this problem has not been well-discussed in the paper. In FL, with large number of clients, i.e. the dimension of q also grows. I am concerned about the efficiency of solving this subproblem while other baselines such as FedAvg/FedProx do not need to have this additional step.
- Keeping the algorithm in the appendix reduces the clarity of the paper.

Questions:
- As FedAvg does not work well with heterogeneous data, I wonder whether FedDrop also suffers in such setting as the update of FedDrop is very similar to FedAvg.
- Also, do we expect the same convergence guarantee as FedAvg?

Minor comment:
- There should be square in equation (6) as it is MSE not RMSE.
- In theorem 2, there should be description for the dot product, typo at \Delta \theta^n_j

**Summary Of The Paper:**

The paper proposes a new method where local workers will drop part of their model using a shared dropout probability received from the server at each communication round. The dropout probabilities are computed by solving an optimization to promote similarity across agent's update. Compared with popular baselines in federated learning, the proposed method demonstrates better performance in terms of number of FLOPs.

**Summary Of The Review:**

I believe the paper contain notable contribution in designing new algorithm using dropout. Numerical experiments illustrate the practical performance of the proposed algorithm However, the proposed lacks discussion on the convergence of the algorithm while other methods like FedAvg/FedProx do have guarantee. The proposed methods appear to only work directly with neural network models, there should be more discussion on other types of model.

---

> ### Author Response · Authors · 2021-11-15
> **Thanks for the detailed review, concerns addressed below.**
>
> First, we would like to address the reviewer’s concerns related to the paper:
> 1. For the concern related to models that are not neural networks, we would like to point out that FedDrop proposed in this paper was specifically proposed for adjusting the dropout probabilities of structured patterns (e.g. channel neurons). We did not design the method with non-neuron-network-based algorithms in mind, and we are open to suggestions on how it can be applied to other learning algorithms.
> 2. For the concern related to discussion of convergence, it is notable that empirically, we did not observe diverging behavior when applying FedDrop on FedAvg. For theoretical analysis, previous works on the convergence of FL algorithms require assumptions on the convexity of the learned model (e.g. for FedAvg [1]). We believe the substantial challenges that come with convergence analysis of NNs are not relevant to the core contribution of FedDrop.
> 3. We would like to point out that Appendix D.2 discussed in detail the computational overhead of the optimization objective. We believe this subsection can address your concerns regarding the overhead and scalability of FedDrop. To summarize:
> 	1. In our test case (Appendix D.2), clients consume 1.29 T FLOPs per round (50% of FedAvg under $r = 0.5$), and the server overhead is ~2% of the total computation.
> 	2. The clients would consume multiple hours of training in a realistic implementation on an iPhone 12 Pro (extrapolating from the profiling result in Table 9), whereas the server side only requires minutes to complete on an Nvidia V100 GPU. This provides a strong case for trading a fraction of server computation for much reduced client computation.
>
> We would like to answer your questions below:
> 1. FedDrop is designed with data heterogeneity in mind, as it optimizes the dropout probabilities of different neurons in multiple clients, allowing the algorithm to concentrate training efforts to neurons that are correlated to the current data distribution of the client. In addition, in Appendix F.2 and Figures 13 and 14, it can be observed that with an increase of data heterogeneity, FedDrop can gain significantly when compared to other FL algorithms.
> 2. Please see 2 above for a discussion of convergence.
>
> We appreciate the reviewer for pointing out the typos in the paper and we will update the paper to reflect the suggestions.
>
> [1]: Li et al., On the Convergence of FedAvg on Non-IID Data, ICLR 2020.

---

### Official Review · Reviewer_tPTk · 2021-11-02

**Correctness:** 3
**Technical Novelty And Significance:** 4
**Empirical Novelty And Significance:** 4
**Recommendation:** 5
**Confidence:** 3

**Main Review:**

Pros:
1. This similarity based dropout technique to reduce communication cost in FL is new, and the authors proposed novel algorithm to approach it.
2. They also did sufficient experiments to empirically validate their method.

Cons:
1. I think the writing and organization can be improved since I am not very clear about some technical details, which I will talk about below.

Questions:

1. I am confused about  $\boldsymbol{p}$ and  $\boldsymbol{p}_c$. I was wondering for $\boldsymbol{p}$, do we optimize a single $\boldsymbol{p}$ or for each client c, we compute a  $\boldsymbol{p}_c$?

2. It seems that the constraint $g(\boldsymbol{p})$ is a combinatorial function since it contains the number of channels. How do you optimize on it?

3. For the experiments, I expect to see FedDrop can converge faster than FedAvg or SCCAFOLD in terms of the wall clock time since the main advantage of it is to reduce model size and hence speedup training.

**Summary Of The Paper:**

This paper proposed a dropout based method towards communication efficient federated learning, where each client will compute the dropout probability based on the update similarity between local models. The experiments show that with fewer FLOPs than traditional federated learning algorithm, FedDrop can achieve the same accuracy.

**Summary Of The Review:**

Overall, the idea of this paper is new, but I think this paper can be better organized to make reader easier to understand the technical details. If the authors can address my questions, I am willing to change my score.

---

> ### Author Response · Authors · 2021-11-15
> **Thanks for the in-depth review, concerns addressed below.**
>
> To address the cons mentioned in the main review, we would like to answer your specific questions:
> 1. The term $\mathbf{p} \in \lvert \mathbf{C} \rvert \times \mathbf{N}$ is a matrix that comprises the dropout keep probabilities of each channel neuron in each client. Following this, $\mathbf{p}_c$ denotes a $\mathbf{N}$-element vector which comprises the dropout keep probabilities of all channel neurons in a client $c \in \mathbf{C}$. We will update the main text accordingly to provide a definition of $\mathbf{p}$.
> 2. The constraint $g(\mathbf{p})$ is not a combinatorial function as the function $\mathrm{flops}(\hat{f}, \mathbf{p}_c)$ evaluates the expected total number of FLOPs of all clients in a training step in terms of the probabilities $\mathbf{p}$, and is thus continuous and differentiable w.r.t.  $\mathbf{p}$. Equation (10) in Appendix B provides a definition for $\mathrm{flops}(\hat{f}, \mathbf{p}_c)$ and it shows that we use the sum of a layer's dropout keep probabilities to estimate the number of active input and output channels.
> 3. FedDrop can provide a faster wall clock time when compared against other methods that employ dense models. As shown in Table 9, we conducted the training on iPhone 12 Pro and found that sparsified models indeed train faster than dense models. In our experimental results, however, we opted to use the number of FLOPs as a metric for comparing computational costs as it is platform and implementation agnostic.
>
> We would like to thank the reviewer for your comments, and will continue to improve the organization of the paper.

---

### Author Response · Authors · 2021-11-30
**We appreciate it if you can kindly check out our responses.**

Dear Reviewers,

Could you kindly check out our responses to the questions.
Hopefully, the answers should address your concerns.
Please let us know if you have any further questions and we will respond to them promptly.

Thank you!


Best regards,

Paper2175 Authors

---

### Decision · Program_Chairs · 2022-01-20

**Decision:**

Reject

**Comment:**

The paper introduces drop-out probabilities which are adaptive to the similarity of model parameters between clients.
The reviewers liked the idea, however missed several aspects, such as a convergence analysis or at least discussion, as well as an analysis of additional cost of the adaptive step, and finally several concerns on the strength of the experimental setup and benchmarks.

Unfortunately consensus among the reviewers is that it remains below the bar even after the discussion phase.

We hope the detailed feedback helps to strengthen the paper for a future occasion.